# Low-volatility compounds contribute significantly to isoprene SOA under high-NO$_x$ conditions

Rebecca H. Schwantes[1,5], Sophia M. Charan[2], Kelvin H. Bates[2,6], Yuanlong Huang[1], Tran B. Nguyen[3], Huajun Mai[1], Weimeng Kong[2], Richard C. Flagan[2], and John H. Seinfeld[2,4]

[1]Division of Geological and Planetary Sciences, California Institute of Technology, 1200 East California Boulevard, Pasadena, California 91125, United States
[2]Division of Chemistry and Chemical Engineering, California Institute of Technology, 1200 East California Boulevard, Pasadena, California 91125, United States
[3]Department of Environmental Toxicology, University of California - Davis, Davis, California 95616, United States
[4]Division of Engineering and Applied Science, California Institute of Technology, Pasadena, California 91125, USA
[5]Current Affiliation: National Center for Atmospheric Research, Boulder, Colorado, 80307, USA
[6]Current Affiliation: Faculty of Arts and Sciences, Harvard University, Cambridge, MA 02138, USA

*Correspondence to:* Rebecca H. Schwantes (rschwant@ucar.edu)

**Abstract.** Recent advances in our knowledge of the gas-phase oxidation of isoprene, the impact of chamber walls on secondary organic aerosol (SOA) mass yields, and aerosol measurement analysis techniques warrant re-evaluating SOA yields from isoprene. In particular, SOA from isoprene oxidation under high-NO$_x$ conditions forms via two major pathways: 1) low-volatility nitrates and dinitrates (LV pathway) and 2) hydroxymethyl-methyl-$\alpha$-lactone (HMML) reaction on a surface or

the condensed phase of particles to form 2-methyl glyceric acid and its oligomers (2MGA pathway). These SOA production pathways respond differently to reaction conditions. Past chamber experiments generated SOA with varying contributions from these two unique pathways, leading to results that are difficult to interpret. This study examines the SOA yields from these two pathways independently, which improves the interpretation of previous results and provides further understanding of the relevance of chamber SOA yields to the atmosphere and regional/global modeling. Results suggest that low-volatility nitrates

and dinitrates produce significantly more aerosol than previously thought; the experimentally measured SOA mass yield from the LV pathway is ∼0.15. Sufficient seed surface area at the start of the reaction is needed to limit the effects of vapor wall losses of low-volatility compounds and accurately measure the complete SOA mass yield. Under dry conditions, substantial amounts of SOA are formed from HMML ring-opening reactions with inorganic ions and HMML organic oligomerization processes. However, the lactone organic oligomerization reactions are suppressed under more atmospherically relevant humidity levels,

where hydration of the lactone is more competitive. This limits the SOA formation potential from the 2MGA pathway to HMML ring-opening reactions with water or inorganic ions under typical atmospheric conditions. The isoprene SOA mass yield from the LV pathway measured in this work is significantly higher than previous studies have reported suggesting that low-volatility compounds such as organic nitrates and dinitrates may contribute to isoprene SOA under high-NO$_x$ conditions significantly more than previously thought, and thus deserve continued study.

# 1 Introduction

In the atmosphere, submicrometer particulate matter is composed of a significant fraction of organic aerosol (Zhang et al., 2007). There are two forms of organic aerosol: primary, which is directly emitted into the atmosphere, and secondary, which is formed when gas-phase compounds partition to the particle phase. Processes governing secondary organic aerosol (SOA) formation are particularly complex (Kroll and Seinfeld, 2008; Hallquist et al., 2009). SOA yields, the ratio of the mass of SOA formed to the mass of the parent volatile organic compound (VOC) reacted, are measured in environmental chambers and are used in models to reduce the complexity of SOA formation.

Isoprene is the dominant non-methane biogenic VOC emitted into the atmosphere. Because of the large flux of isoprene ($\sim$535 Tg yr$^{-1}$) into the atmosphere (Guenther et al., 2012), oxidation of isoprene is a significant source of SOA even though SOA yields measured in chambers are relatively low (Carlton et al., 2009). Despite numerous experimental studies of isoprene SOA formation under varying conditions (Pandis et al., 1991; Edney et al., 2005; Kroll et al., 2005; Dommen et al., 2006; Kleindienst et al., 2006; Kroll et al., 2006; Ng et al., 2006; Paulot et al., 2009; Chan et al., 2010; Chhabra et al., 2010; Surratt et al., 2010; Nguyen et al., 2011; Zhang et al., 2011, 2012; Lin et al., 2013; Nguyen et al., 2014b; Xu et al., 2014; Krechmer et al., 2015; Lambe et al., 2015; Nguyen et al., 2015; Bregonzio-Rozier et al., 2015; Clark et al., 2016, etc.), a consensus on the magnitude of SOA formed from isoprene oxidation by the hydroxyl radical (OH) is still lacking (Carlton et al., 2009; Bregonzio-Rozier et al., 2015; Clark et al., 2016). This lack of consensus in the experimental data leads recent global modeling studies (Marais et al., 2016; Stadtler et al., 2018) to implement SOA schemes that produce significantly different overall isoprene SOA yields. Isoprene SOA yields have been shown to depend on a variety of factors including RO$_2$ fate, NO$_2$/NO ratio, relative humidity, degree of oxidation, temperature, seed surface area, particle acidity, and chamber irradiation source (Carlton et al., 2009). These experimental conditions have not always been controlled or reported, which is likely a major reason for the variability seen in past isoprene SOA yields. By measuring isoprene SOA yields while controlling for seed surface area, RO$_2$ fate, NO$_2$/NO ratio, relative humidity, and temperature, we seek to resolve uncertainties in SOA formation in past yields.

Recent advances have improved our understanding of how chamber SOA yields should be measured and analyzed. This includes accounting carefully for particle wall deposition (Loza et al., 2012), vapor wall deposition (Zhang et al., 2014; Ehn et al., 2014), and particle coagulation (Nah et al., 2017). Advances have also taken place in the data processing of aerosol size distribution measurements by the differential mobility analyzer coupled to a condensation particle counter (DMA-CPC), the main instrument used to measure SOA yields (Mai and Flagan, 2018; Mai et al., 2018). Because isoprene SOA yields tend to be relatively small, the DMA data inversion technique and correction for CPC response time are quite important.

Additionally, there have been major recent advances in our understanding of isoprene gas-phase oxidation (Wennberg et al., 2018, and references therein) including theoretical (e.g., Peeters et al., 2009, 2014; Kjaergaard et al., 2012) and experimental (e.g., Teng et al., 2017; Nguyen et al., 2015; Lee et al., 2014; Jacobs et al., 2014) studies. This improved understanding of isoprene gas-phase chemistry influences the processes governing isoprene SOA formation and informs the experimental design of the present work. This work focuses on the production of SOA from OH-initiated isoprene oxidation under high-

$NO_x$ conditions, which occurs via two major chemical pathways (Figures 1 and 2). The first we define throughout as the low volatility (LV) pathway representing all aerosol formed from the equilibrium gas/particle partitioning of compounds with sufficiently low volatility, which mostly include functionalized nitrates and dinitrates (e.g., red compounds in Figure 1). The second we define as the 2-methyl glyceric acid (2MGA) pathway representing aerosol formed from 2MGA, its oligomers, its organosulfates, and its organonitrates (blue compounds in Figure 2). There are many definitions for high-$NO_x$ conditions (Wennberg, 2013). Here we test two different high-$NO_x$ chemical regimes. Experiments targeting the LV pathway are designed such that all peroxy radicals including acyl peroxy radicals dominantly react with NO and experiments targeting the 2MGA pathway are designed such that all acyl peroxy radicals dominantly react with $NO_2$ and all other peroxy radicals dominantly react with NO.

Aerosol from the LV pathway is believed to be composed largely of isoprene dihydroxy dinitrates, which are produced from the first-generation hydroxy nitrate reacting with OH to form a peroxy radical that then reacts with NO. The gas-phase yield of isoprene dihydroxy dinitrates is quite uncertain (Lee et al., 2014). In general, the nitrate yields from highly functionalized $RO_2$ radicals have not been well studied (Wennberg et al., 2018) due mostly to difficulties in measuring such low volatility compounds. The formation of some organic nitrate SOA-precursors are summarized in Figure 1, which is largely adapted from schemes presented in Wennberg et al. (2018) with the exception of the isoprene dihydroxy nitrooxy alkoxy radical 1,5 H-shift. Wennberg et al. (2018) suggests the importance of a similar peroxy radical 1,5 H-shift, which will not form in the present experiments due to the high levels of NO. However, based on past studies largely on alkane oxidation (Orlando et al., 2003; Atkinson, 2007), the equivalent alkoxy radical 1,5 H-shift is expected to occur and has the potential to form low-volatility nitrates as further described in Section 5.1.

Throughout the text we use low-volatility as a general term representing gas-phase compounds with a potential to exist partially in the particle phase. In this work, low-volatility compounds include the following volatility classes from Donahue et al. (2012): IVOC (Intermediate), SVOC (Semi-), LVOC (Low), and ELVOC (Extremely Low). When referring to specific volatility classes, the acronyms defined above are used.

Aerosol from the 2MGA pathway forms when methacrolein is oxidized under high-$NO_2$ conditions to form methylacryloyl peroxynitrate (MPAN). MPAN reacts with OH to form hydroxymethyl-methyl-$\alpha$-lactone (HMML), and HMML either decomposes in the gas phase to form hydroxy acetone or interacts with a wet surface to form 2-MGA (Kjaergaard et al., 2012; Nguyen et al., 2015). A minor channel to form methacrylic acid epoxide (MAE) also exists from methacrolein oxidation (Lin et al., 2013), but not from pure MPAN oxidation (Nguyen et al., 2015). Nguyen et al. (2015) demonstrated that MAE does not easily undergo ring-opening reactions to form particles. Thus, the yield of MAE from MACR oxidation reported in Lin et al. (2013) should be adjusted to include only MAE detected in the gas phase, which corresponds to a yield of $\sim$1-2%.

Because SOA formed from the LV and 2MGA pathways is chemically distinct in both route of formation and composition, the experiments reported here probed these chemical pathways separately. This experimental design is aimed to resolve inconsistencies associated with previously reported isoprene SOA yields (Carlton et al., 2009) and improve our understanding of isoprene SOA formation. Additionally, we seek to report updated isoprene SOA yields as well as trace the SOA yields to known gas-phase SOA precursors.

## 2 Experimental Methods

Chamber experiments were performed to study SOA formation from isoprene oxidation under high-$NO_x$ conditions from two distinct pathways: 1) low-volatility nitrates and dinitrates (LV pathway) and 2) 2-MGA and its oligomers (2MGA pathway). Experiments targeting the LV pathway were performed using isoprene as the precursor and an $NO_2$/NO ratio < 1.5 was maintained throughout the entire experiment (as verified by the kinetic mechanism) in order to favor the formation of nitrates and dinitrates and limit the formation of MPAN (Figure 1). Experiments targeting the 2MGA pathway were performed using methacrolein as the precursor and an $NO_2$/NO ratio > 11 was maintained throughout the entire experiment (as verified by the kinetic mechanism) with the exception of experiment M9, which maintained an $NO_2$/NO ratio > 8. This high $NO_2$/NO ratio accentuated the formation of MPAN, and thereby 2MGA (Figure 2) and was important for reducing variability between the experiments. If a lower $NO_2$/NO ratio was used, small fluctuations in the initial $NO_2$ or NO would result in large differences in the $NO_2$/NO ratio. In order to completely separate the LV and 2MGA pathways, methacrolein had to be used as the VOC precursor for the 2MGA pathway experiments. If isoprene was used, even at the high $NO_2$/NO ratios used in the 2MGA pathway experiments, the SOA precursors from the LV pathway would form resulting in a mixed regime (i.e., chemistry in Figure 1 is not dependent on $NO_2$ concentration). In each case, the effect of seed surface area, temperature, and humidity on the SOA yield was independently determined.

### 2.1 Experimental Conditions

Experiments (see Table 1) were conducted in the Caltech dual chamber facility using a 21 m$^3$ Teflon chamber. Prior to each experiment, the chamber was flushed with dry, purified air for 24 h. For humid experiments, the chamber was humidified prior to all injections. 25°C ultrapure water (18 M$\Omega$, Millipore Milli-Q) was recirculated through a Nafion membrane humidifier (FC200, Permapure LLC) while purified air was flowed through the humidifier and into the chamber. First, isoprene (99% purity) or methacrolein (95% purity) was injected into a glass bulb using a gas-tight syringe and was carried by a flow of dry nitrogen into the chamber.

Second, methyl nitrite ($CH_3ONO$) was injected into the chamber. $CH_3ONO$ was synthesized using the technique described in Taylor et al. (1980) and Chan et al. (2010) and stored in liquid nitrogen. Prior to each experiment, an evacuated glass bulb was filled with $CH_3ONO$ to the desired pressure, as measured by a capacitance manometer (MKS Baratron$^{TM}$). This bulb was then backfilled with nitrogen and flushed into the chamber. The bulb pressure was used to calculate the $CH_3ONO$ mixing ratio in the chamber (see Table 1). After $CH_3ONO$ was injected, pulses of purified air were added to the chamber to enhance mixing. Once the chamber was adequately mixed, NO (501 ppm in $N_2$, Scott Specialty Gases) or $NO_2$ (488 ppm in $N_2$, Scott Specialty Gases) was injected into the chamber through a calibrated mass flow controller. Again the chamber was mixed by pulses of purified air.

Seed particles were generated from an atomizer using 0.06 M $(NH_4)_2SO_4$ seed solution. The seed aerosol was directed through a soft x-ray neutralizer (TSI Model 3088) prior to injection into the chamber to ensure a consistent initial particle charge distribution for all experiments. For humid experiments, the seed aerosol was directed through a wet-wall denuder after

exiting the neutralizer in order to ensure the particles were deliquesced. After seed injection, mixing air was turned on for 1 min to enhance mixing. The seed aerosol particle number concentration had an approximately lognormal diameter distribution centered on average $\sim$100 nm.

After injecting all gas-phase precursors and seed aerosol, photooxidation was delayed by 1 h for experiments with no initial seed aerosol and 4 h for experiments with initial seed aerosol. The rate of particle wall deposition was measured for each experiment during this 4 h delay. Although $NO_2$ was not intentionally added for the LV pathway experiments, a modest $NO_2$ signal was observed to form during the 4 h delay and is reported in Table 1. This "$NO_2$" signal may be $NO_2$ itself or an interference in the $NO_x$ monitor from an $NO_y$ compound (e.g., known interferences include organic nitrates, nitrous acid, and $CH_3ONO$). The small signal of $NO_2$, or other $NO_y$ compound, is not expected to influence the results given the significantly larger initial NO levels (Table 1). When $NO_2$ or $CH_3ONO$ were injected into the chamber, an NO signal was observed on the $NO_x$ monitor. As the $NO_x$ monitor has few interferences for NO, a small fraction of NO was likely formed from $NO_2$ or $CH_3ONO$ photolysis in the Teflon injection lines. Thus, the slight increase of NO with the $NO_2$ and $CH_3ONO$ injection was assumed and reported to be initial NO (Table 1).

The Caltech chamber uses Ultraviolet (UV) broadband lights with the main emission peak centered at $\sim$350 nm. Only 10% of full light capacity ($j_{NO2\ 10\%}$ = 4 x $10^{-4}$ $s^{-1}$) was used for these experiments because $CH_3ONO$ photolyzes rapidly, and the lower light intensity minimizes chamber temperature increases ($\sim$0.4 °C on average) caused by the UV lights. In all experiments, OH was produced by the photolysis of $CH_3ONO$ as shown in the following reactions:

$$CH_3ONO + h\nu \rightarrow CH_3O + NO \tag{R1}$$

$$CH_3O + O_2 \rightarrow CH_2O + HO_2 \tag{R2}$$

$$HO_2 + NO \rightarrow OH + NO_2 \tag{R3}$$

Relative to other OH precursors, $CH_3ONO$ has a low Henry's law constant (15 M $atm^{-1}$, calculated by theory) (Sander, 2015). During experiments with high RH, unlike other OH precursors, $CH_3ONO$ is not expected to enhance OH production in the particle phase beyond atmospherically relevant levels.

## 2.2 Instrumentation

Temperature and relative humidity (RH) were measured using a Vaisala HMM211 probe. NO and $NO_2$ were monitored using a Teledyne $NO_x$ analyzer (T200). Because the Teledyne $NO_x$ monitor detects $CH_3ONO$, organic nitrates, and other $NO_y$ compounds as $NO_2$, only initial $NO_2$ measurements can be constrained with this instrument. For some experiments, $NO_2$ was also monitored using a luminol $NO_2$/acyl peroxynitrate analyzer developed by Fitz Aerometric Technologies. This instrument separates $NO_2$ via chromatography at room-temperature using a deactivated DB-5 column. $NO_2$ then reacts with luminol to

produce a chemiluminescence response (Gaffney et al., 1998). The $NO_2$ measured by the luminol $NO_2$/acyl peroxynitrate analyzer compares reasonably well with the simulated $NO_2$ from the kinetic model (Figure S3).

A gas chromatograph with a flame ionization detector (GC-FID, HP 6890N, column HP-Plot-Q) was used to measure the decay of isoprene and methacrolein. The GC-FID was calibrated with ~50-60 ppm of isoprene or methacrolein generated from analytical standards (Aldrich 95-99% purity) and cross-calibrated by fourier transform infrared absorption (FT-IR) spectroscopy (pathlength 19 cm) using the absorption cross sections measured by Pacific Northwest National Laboratory (PNNL) for isoprene or methacrolein (Sharpe et al., 2004). Linearity in the GC-FID calibration was determined to an error of ~1% across a factor of 150 in dilution.

Aerosol organic and inorganic composition was recorded in-situ using a high-resolution time-of-flight aerosol mass spectrometer (HR-AMS, Aerodyne Research, Inc.). The HR-AMS switched every 1 min between the high-resolution W-mode and the lower-resolution, higher-sensitivity V-mode. The data were analyzed with Igor Pro (Wave Metrics, Inc.), utilizing the Squirrel 1.56D and PIKA 1.15D analysis toolkits (from http://cires1.colorado.edu/jimenez-group/ToFAMSResources/ToFSoftware/index.html). In-line filter runs conducted prior to each experiment were used to correct for air interferences (Aiken et al., 2008). Bulk SOA elemental composition was calculated following the methods and recommendations of Aiken et al. (2008) and Canagaratna et al. (2015).

Aerosol volume and number concentration were monitored using a differential mobility analyzer (DMA, TSI 3081 column) coupled with a condensation particle counter (CPC, TSI 3010), which measures all particles with a diameter between 20-800 nm. The voltage scan used by the DMA was 1 min hold at 15 V, 4 min increase to 9850 V, 1 min hold at 9850 V, and 0.5 min decrease back to 15 V. Only the upscan data were used for the analysis. The longer upscan and hold times used here, compared to previous studies (e.g., Loza et al. (2012) and Zhang et al. (2014)), reduced biases caused by mixing in the CPC. Such CPC mixing biases particularly impact the measurement of large particles, which are important in SOA yield experiments as they contribute significantly to the total SOA volume.

The DMA data analysis includes an improved data inversion and a correction for particle mixing in the condensation particle counter, which influences the CPC response time (Mai and Flagan, 2018; Mai et al., 2018). The inversion technique is applicable only to particles $\leq$ 600 nm. Particle concentration between 600-800 nm was calculated assuming a non-linear least squares lognormal fit applied to particles from 400-600 nm. For experiments with no initial seed aerosol, the inversion inconsistently determined the presence of particles beyond 400 nm. Such large particles are unlikely to be the result of nucleation and more likely to represent an artifact of the inversion; thus, only particles <400 nm were used in the analysis of these experiments. Corrections to the DMA data for coagulation and particle wall loss are addressed in Section 4.1.

Isoprene oxidation products were measured using a $CF_3O^-$ chemical ionization mass spectrometer (CIMS), which utilizes a custom-modified triple quadrupole mass analyzer (Varian 1200) (St. Clair et al., 2010). $CF_3O^-$ interacts with a gas-phase compound (A) to form a complex that is detected at the molecular weight of A + 85 or, in some cases, to fragment. Various fragmentation products can form as explained in previous work (e.g., Paulot et al. (2009); Praske et al. (2015); Schwantes et al. (2017)). In this work, the CIMS results are only used to identify the presence of highly functionalized organic nitrates and not for quantification, so only signals from the complex (i.e., A • $CF_3O^-$) and not from fragmentation are reported.

## 3 Kinetic Mechanism

All relevant reactions included in the Master Chemical Mechanism (MCM) v3.3.1 (http://mcm.leeds.ac.uk/MCM) were used in the current kinetic model (Jenkin et al., 1997; Saunders et al., 2003). Isoprene oxidation chemistry was recently updated in MCM v3.3.1 by Jenkin et al. (2015). Additional reactions included in the kinetic model, but not in MCM v3.3.1 are listed in Table S1. Updates include inorganic reactions needed for chamber studies with large $NO_x$ levels (e.g., $CH_3ONO$ photolysis) and small changes to the isoprene chemistry based largely on Wennberg et al. (2018) and consistent with Figures 1 and 2. As shown in Table S1, these updates include: the first generation isoprene hydroxy nitrate yields, the rates and branching ratios for the oxidation of the first generation isoprene hydroxy nitrates, and the HMML yield from the MPAN + OH reaction. In some cases, $\delta$-isoprene hydroxy alkoxy radicals in MCM v3.3.1 decompose through peroxy radical H-shifts directly to products that would not form under the high-NO conditions in this work. For simplicity, we change these reactions, so that the $\delta$-isoprene hydroxy alkoxy radicals form unity yields of hydroxy aldehydes. BOXMOX, a box-model software package using the Kinetic PreProcessor (Knote et al., 2015), was used to simulate the chamber experiments. As listed in Table 1, the kinetic model was initialized for each experiment with the measured initial concentration of VOC, NO, $NO_2$, and $CH_3ONO$ as well as the measured average temperature and relative humidity.

Saturation mass concentration (C*) and fraction of each compound in the particle phase ($F_P$) at 13, 26, and 32°C are estimated for relevant organic nitrates and dinitrates produced in MCM v3.3.1 and listed in Table S2. C* was calculated with the vapor pressure estimated from Nannoolal et al. (2004, 2008) using the online calculator located at: http://www.aim.env.uea. ac.uk/aim/ddbst/pcalc_main.php. $F_P$ was calculated from the C* values and gas-particle equilibrium theory as further explained in Section S1 (Seinfeld and Pandis, 2016).

As shown in Table 1, the inferred OH concentration was larger in experiments with higher temperatures. Because the temperature dependence of the $CH_3ONO$ absorption cross section and quantum yield are not well established, the $CH_3ONO$ photolysis rate constant was calculated from the $CH_3ONO$ decay curve as measured by the GC-FID. Unfortunately, the GC-FID sensitivity to $CH_3ONO$ was low, so only the 2MGA experiments produced a sufficiently large signal for this approach. The average $CH_3ONO$ photolysis rate constant from experiments M1-M3, M4, M5, and M6-M8 were used for dry $\sim$25°C (1.9 x $10^{-4}$ s$^{-1}$), dry $\sim$13°C (1.4 x $10^{-4}$ s$^{-1}$), dry $\sim$32°C (2.3 x $10^{-4}$ s$^{-1}$), and humid $\sim$25°C (1.9 x $10^{-4}$ s$^{-1}$) experiments, respectively. This approach accurately captured the reaction of isoprene and methacrolein with OH in all experiments (Figures S1 and S2), which implies that the simulated OH in the kinetic model is reasonably accurate even over varying temperature. All other photolysis rate constants are calculated from the absorption cross sections and quantum yields reported in Burkholder et al. (2015) and Jenkin et al. (2015). Additionally, the kinetic model captures NO and $NO_2$ reasonably well for both the LV and 2MGA pathway experiments (Figure S3).

## 4 Results

First, corrections for particle coagulation and particle wall deposition, which are required for accurate calculation of SOA yields, are addressed (Section 4.1). Next, SOA produced from the LV (Section 4.2) and 2MGA (Section 4.3) pathways are discussed.

### 4.1 Corrections for Particle Coagulation and Particle Wall Deposition

Past studies reporting particle wall deposition coefficients apply the measured particle number decay rate in each size bin to produce a wall deposition coefficient, ($\beta(D_p,t)$), that is a function of particle size ($D_p$) (Loza et al., 2012). Because larger seed particle number and surface area concentrations were used in these experiments, corrections to $\beta(D_p,t)$ that account for coagulation are needed (Pierce et al., 2008; Nah et al., 2017). The current work uses an approach similar to that of Nah et al. (2017) and Sunol et al. (2018) with updates to account for electrostatic charges on the chamber walls as described by Charan et al. (2018). To reduce the experimental uncertainty associated with these processes, particle wall deposition was calculated during each experiment. This approach accounted for the day-to-day fluctuations in particle coagulation processes, chamber wall charging, and chamber mixing. We summarize these approaches and describe any changes required for this analysis in Section S2 of the Supplement.

Four particle wall loss experiments were performed under dry conditions (RH <10%) at varying seed surface areas as controls to verify the technique used to correct for particle wall loss, particle coagulation, and electrostatic charges on the chamber walls. These particle wall deposition experiments were performed by injecting ammonium sulfate seed into the chamber, as described in Section 2.1. Mixing air was added, and the ammonium sulfate seed aerosol was monitored in the dark chamber for at least 14 h. These controls confirmed that the wall loss correction calculated over the first 3.5 h could be extrapolated for an additional 10 h. Beyond 10 h, the wall loss correction was more uncertain, so results only from the first 10 h of each experiment are reported. The percent change between the aerosol volume over 10 h and the aerosol volume at the start of the control experiment was between +4% and -6% for all dry control experiments (Figure 3).

The results of these control experiments verified the robustness of the correction technique and provided an estimate for the uncertainty. The reported uncertainty for the particle wall deposition correction is +4% and -6% of the corrected aerosol volume at the start of photooxidation. Experiments with larger seed aerosol volumes exhibit larger uncertainty in the reported SOA yield. However, such experiments are necessary despite the extra uncertainty, as larger seed surface areas minimize low biases in SOA yields due to vapor wall deposition of low-volatility compounds (Zhang et al., 2014; Ehn et al., 2014). For experiments with no initial seed aerosol, particle wall loss corrections were applied assuming the particles coagulated and deposited similarly to the lowest aerosol loading control experiment (C1). No uncertainty for the particle wall deposition correction was added to these experiments because the uncertainty derived here is applicable only to experiments with initial seed aerosol.

In experiments C1-C4, D1-D9, and M1-M8, electrostatic charges on the chamber walls were inferred to be present. After these experiments were completed, new Teflon chambers were acquired with negligible electrostatic charges on the chamber

walls (Charan et al., 2018) likely due to their smaller volume (18 m$^3$). Three additional new experiments (D10, D11, and M9) were completed using one of these new Teflon chambers to confirm that we had accurately corrected for the chamber wall charging effects. For the LV pathway experiments (Section 4.2), results for the new experiments were quite similar and within uncertainties of the old experiments. The new 2MGA pathway experiment produced slightly lower SOA yields than the old experiments, but not necessarily because of the chamber wall charging corrections as described in Section 4.3.

Five control experiments were also performed under humid conditions. The DMA cannot measure hydrated particles owing to arcing in the DMA column at high RH. Thus, a Nafion dryer was used to dry particles before measurement. For the coagulation correction, the volume of the hydrated seed was calculated based on the dry DMA particle measurement, the RH in the chamber, and the hygroscopic growth curve for ammonium sulfate measured by Sjogren et al. (2007). The percent change for the aerosol volume was higher and less consistent in the humid control experiments than in the dry control experiments. Also, the optimized value of the electric field ($\bar{E}$) was higher in many of the humid experiments than in the dry experiments (Section S2, Table S3). Increased humidity is expected to decrease the electrostatic charges on the chamber walls (e.g., Ribeiro et al. (1992)), but the inferred $\bar{E}$ suggests the opposite. Possibly, the humidifying process enhanced the electrostatic charges on the chamber walls, or nitric acid, which is enhanced in the particle phase in the humid experiments under high-NO$_x$ conditions, impacts the coagulation or particle wall loss processes.

The AMS data confirm that during the humid experiments, nitric acid partitioned to the particle phase and that organic aerosol was produced during photooxidation for all experiments. Nevertheless, the particle wall loss corrected volume measured by the DMA decayed below zero during photooxidation in the humid experiments. Potentially, this DMA volume decay suggests that nitric acid present in the particle phase changes the particle coagulation or wall loss characteristics. Even if we understood the impact of nitric acid on the particle coagulation or wall loss corrections, assessing how much of the particle growth is due to nitric acid versus organics would be difficult with the DMA, which measures only total aerosol volume and not composition. Further chamber characterization is required in order to assess isoprene SOA yields measured by the DMA from humid experiments under high-NO$_x$ conditions. Thus, in this work, only the AMS results will be discussed for the humid experiments and SOA yields are only reported for experiments performed under dry conditions (Table 1). None of the dry experiments exhibited the odd behavior observed in the humid experiments, and the AMS results confirm that under dry conditions minimal nitric acid partitioned to the aerosols (Figure S10). For the dry experiments, the uncertainties are well characterized by the dry control experiments presented in Figure 3.

## 4.2 SOA Formation From the LV Pathway

The SOA mass yields from isoprene for all LV pathway experiments (i.e., experiments targeting low-volatility compounds) are shown in Figures 4 and 5. To convert aerosol volume measured by the DMA to aerosol mass, a density of 1.4 g cm$^{-3}$ was assumed, consistent with past work (Dommen et al., 2006; Kroll et al., 2005, 2006; Bregonzio-Rozier et al., 2015). The kinetic mechanism suggests that in all experiments targeting the LV pathway, formation of HMML was < 0.12 ppb even in experiments performed under cold conditions (13°C). The AMS results also confirm that 2MGA and its oligomers are not present in the LV

pathway experiments (Section 5.2). Thus, the kinetic mechanism and AMS results verify that the experimental design correctly separates the two chemical regimes and 2-MGA is not substantially adding to the aerosol mass in the LV pathway experiments.

Aerosol growth in the absence of seed aerosols was not observed in the LV pathway experiments (Figure 4). As expected, SOA formed from gas/particle partitioning in the LV pathway exhibited a large dependence on seed surface area (Figure 5). Without sufficient seed aerosol, low-volatility nitrates partition primarily to the chamber walls, and the resulting SOA yields are biased low. With the addition of inorganic seed aerosol like ammonium sulfate, vapor species are expected to partition more to particles relative to the chamber wall (Zhang et al., 2014). The gas-particle equilibrium is not expected to be dependent on the concentration of inorganic seed aerosol, but instead is dependent on the concentration of organic aerosol. Depending on the saturation mass concentration (C*), as the concentration of organic aerosol rises, vapors are present more in the particle-phase relative to the gas-phase (Seinfeld and Pandis, 2016). C* and the fraction of a compound expected to be in the particle phase ($F_P$) were estimated for a variety of organic nitrates and dinitrates in MCM v3.3.1 at 13, 26, and 32°C (Table S2).

Similar to previous studies (e.g., Zhang et al. (2014)), at a certain point increased seed surface area no longer substantially impacts the SOA yield (i.e., Figure 5 after 2500 $\mu m^2$ $cm^{-3}$). This point will heavily depend on the system and the saturation mass concentration (C*) of the SOA precursors. As shown in Table S2, the isoprene SOA precursors are mostly classified as IVOCs and SVOCs (Donahue et al., 2012). Reaching a point where most of the vapors are in particles relative to the chamber wall is expected for IVOCs and SVOCs, which have moderate vapor wall losses in Teflon chambers especially under dry conditions (Zhang et al., 2014; Huang et al., 2018).

As expected the isoprene dihydroxy dinitrates had the lowest C* values and high $F_P$ at all temperatures. Based on the kinetic model, even assuming all of the isoprene dihydroxy dinitrates exist in the particle-phase, the SOA formed would be much less than that detected in this study (Figure S4). This is likely caused by too low production of the isoprene dihydroxy dinitrates and/or the importance of other SOA precursors. There are many additional compounds largely produced from hydroxy alde-hyde oxidation or alkoxy [1,5]-H-shifts (Figure 1) with $F_P$ at 26°C between 0.05 and 0.4 (Table S2). The vapor pressures may be over-predicted for these specific compounds, but past studies suggest that in general, vapor pressure estimation methods like Nannoolal et al. (2004, 2008) under-predict rather than over-predict vapor pressure (Kurten et al., 2016). From the C* calculations (Table S2), none of the multi-functional organic nitrates are expected to be appreciably in the particle phase. How-ever, in a recent study, Lee et al. (2016) detected many multi-functional organic nitrates in aerosols in the ambient atmosphere, which has lower organic aerosol concentrations than chamber studies. Possibly, MCM under-predicts the formation of these IVOC and SVOC products (Figure S4), such that even if only a fraction exists in the particle-phase relative to the gas-phase, an appreciable mass of aerosol still forms and/or these results suggest that volatility is not the only driver for aerosol formation from the LV pathway. All of the multi-functional nitrates here with estimated $F_P$ at 26°C between 0.05 and 0.4 have at least one hydroxy or aldehyde group (Table S2). Alcohols and aldehydes are well known to combine in particles to produce hemiacetals, whose vapor pressure is significantly lower than that of the initial reactants (Kroll and Seinfeld, 2008). Several past studies have confirmed that $NO_x$ in general, but not necessarily linearly, decreases the volatility of isoprene SOA (Kleindienst et al., 2009; Xu et al., 2014; D'Ambro et al., 2017). This decrease in volatility is likely due to accretion reactions. Whether the accretion

reactions from hemiacetal formation versus those from 2MGA oligomerization are responsible for the decrease in volatility is yet unknown.

Differences in the SOA yield at 10h of photooxidation by varying temperature (13-32°C) lie within the experimental uncertainty. SOA forms earlier (i.e. with less isoprene reacted) at 13°C than at 26 or 32°C at comparable seed surface areas. This is consistent with the C* values estimated in Table S2 and the above explanation demonstrating the likelihood of accretion reactions. Vapors that are only moderately in the particle phase at 26°C (e.g., $F_P$ = 0.05-0.4) will exist more appreciably in the particle phase at 13°C (e.g., $F_P$ = 0.2-0.8). From the above discussion, we expect many of these compounds are SOA precursors not based only on their volatility, but also due to their potential to react in the particle phase to form lower volatility products such as hemiacetals. Thus, if accretion reactions are the main factor, reducing temperature is expected to increase the rate of SOA production, but not necessarily to impact the overall SOA yield.

Clark et al. (2016) have also measured isoprene SOA formation under high-$NO_x$ conditions at varying temperatures. Under the high-$NO_x$ conditions of their study, SOA is produced from both the 2MGA and LV pathways combined. Similarly to our study, Clark et al. (2016) do not find appreciable differences for temperatures from 27-40°C. Contrary, to our work, Clark et al. (2016) found that reducing the temperature to 5°C increases the SOA yield by a factor of 4. Unfortunately, there are no experiments between 5°C and 27°C to determine whether this shift is exponential or linear, so direct comparison of our results at 13°C is difficult. Under the high-$NO_x$ conditions used by Clark et al. (2016), at colder temperatures MPAN will be more stable and so more HMML will form, which produces more SOA. Under this mixed regime, determining how much of the SOA increase is due to the LV versus the 2MGA pathway for direct comparison to this study is difficult. Additionally, Clark et al. (2016) start with significantly more isoprene than in our experiment, which enhances the concentration of organic aerosol, which will increase the fraction of a compound in the particle phase relative to the gas phase.

While vapor wall losses of LV compounds are expected to increase at colder temperatures (Zhang et al., 2015; Schwantes et al., 2017), the organic nitrate yields are also expected to be enhanced under colder temperatures (Orlando and Tyndall, 2012). Thus, the effects of these two temperature-dependent processes might cancel. The increase in organic nitrate yield is expected to be moderate. For example, ∼30% increase from 32 to 13°C is estimated for the yield of isoprene hydroxy nitrates (Wennberg et al., 2018). The loss of vapors to the walls could be much higher at colder temperatures, but this is hard to constrain as vapor wall deposition is dependent on the compound itself and the chamber used. The chamber used by Clark et al. (2016) (90 m$^3$) is larger than our chamber (21 m$^3$). Vapor wall losses are expected to be lower in larger chambers, which have a lower chamber surface area to volume ratio (Zhang et al., 2015). Significant seed aerosol is added into our chamber to reduce the influence of vapor wall deposition, but vapor wall deposition could certainly explain some of the differences at cold temperatures between our results and those from Clark et al. (2016).

Consistent with past work (e.g., Kroll et al. (2005); Ng et al. (2006)), aerosol from the LV pathway is produced only after most of the isoprene is consumed, implying that aerosol from the LV pathway largely forms from later-generation chemistry (Figure 4). As shown in Figure 4, generally, SOA formation begins earlier (i.e., with less isoprene reacted) in experiments with larger seed aerosol. This is consistent with vapors partitioning more to particles relative to the chamber wall when seed aerosol is enhanced. The extent to which later-generation products are oxidized (i.e., the degree of oxidation) impacts the SOA yield

as demonstrated by the varying slope (i.e., SOA yield) during each experiment in Figure 4. We tested the OH/isoprene ratio on the SOA yield. Experiment D7 was performed with 40 ppb of isoprene compared to 55-60 ppb used in the other experiments, while the OH precursor concentration was kept constant. The kinetic model predicts that the production of important gas-phase SOA precursors from the LV pathway (e.g., isoprene dihydroxy dinitrates), when corrected for total isoprene reacted, is similar in experiment D7 to the other experiments (Figure S4). The empirical results are consistent with these predictions. Although a lower isoprene loading decreases the competition of isoprene with OH, other compounds also react with OH quickly (e.g., NO). Under the conditions used in this study, differences in isoprene loading are not expected to greatly influence the isoprene SOA mass yield. However, detailed kinetic modeling of past experimental conditions would be necessary to understand how the degree of oxidation of later-generation products in this study compares to other studies.

In summary, the results (Figure 5) suggest that one of the most important metrics for understanding the variability in SOA production from the LV pathway in various chamber experiments may be the initial seed surface area, instead of temperature or OH/isoprene ratio. Other parameters such as humidity and seed composition may also be important for SOA yields, but were not tested in this study. Future experiments examining SOA yields should report the initial seed surface area and use a sufficient seed loading to reduce the impact of vapor wall deposition.

## 4.3  SOA Formation from the 2MGA Pathway

The SOA mass yields from methacrolein for all 2MGA pathway experiments (i.e., experiments targeting 2MGA and its oligomers) are shown in Figures 6 and 7. Results from past experiments (Chan et al., 2010) have already demonstrated that fluctuations in the $NO_2$/NO ratio impacts SOA formation through the production of MPAN. In this work, the $NO_2$/NO ratio is kept as consistent as possible to isolate other influences on SOA production. The kinetic model suggests that the conditions for each experiment produce a consistent level of HMML (Figure S4). Interestingly, because the experimental conditions heavily favored MPAN formation, the level of OH available to react with MPAN became the limiting reactant for aerosol formation in each experiment.

Contrary to the LV pathway, SOA in the 2MGA pathway experiments do not require seed particles to form. The process of SOA formation from these two pathways is very different. Lactone SOA precursors may polymerize in the presence of organics and water, which possibly explains why SOA from the 2MGA pathway readily forms particles without significant seed surface area, whereas in the LV pathway experiments volatility-based SOA formation results in aerosol yields that are particularly impacted by vapor partitioning. For the 2MGA pathway experiments, even though SOA formation occurred without initial seed aerosol, larger initial seed loadings still enhanced the SOA yield (Figure 7). Possibly, similar to the LV pathway, larger seed surface areas limit vapor wall loss of HMML or its oligomerization partners. Alternatively, the presence of higher ammonium sulfate seed aerosol may also increase organosulfate formation, which could impact SOA composition and yield.

Temperature was varied between 13°C to 32°C. The $NO_2$/NO ratio used in this work was sufficiently high such that this temperature change did not greatly influence MPAN or HMML formation (Figure S4). Thus, these experiments only test whether aerosol properties and SOA yields are affected by temperature, as MPAN thermal decomposition is minimized. At the high $NO_2$/NO ratios used in this work, temperature does not impact SOA mass yield beyond given uncertainties (Figure 7).

Based on known gas-phase chemistry, past studies (e.g., Clark et al. (2016)) with more moderate $NO_2/NO$ ratios than that used in this work are expected to measure an enhanced SOA yield under colder temperatures due to a reduction in MPAN thermal decomposition and thereby an increase in HMML formation.

HMML, based on volatility alone, would exist mostly in the gas phase, but because HMML is very reactive (e.g., oligomerization or reaction with inorganic ions in the particle phase), HMML quickly produces aerosol (Kjaergaard et al., 2012; Nguyen et al., 2015). Based on HMML production simulated by the kinetic mechanism under the conditions used in these experiments, ∼0.21 SOA mass yield from methacrolein is expected purely from the mass contained in HMML (MW = 102 g/mol, Figure S4). At first, the molecular weight of HMML itself is used because this is the mass of the majority of the oligomer monomers. This represents about half of the SOA mass yield (∼0.5) measured from the experiment performed with the highest seed surface area. The rest of the aerosol is likely comprised of inorganic or organic compounds that react with HMML in the particle phase. For example, inorganic compounds such as water, nitrate, and sulfate can react with HMML through ring-opening reactions to produce total methacrolein SOA mass yields of ∼0.25, ∼0.34, and ∼0.41, respectively (Figure 2). Additionally, HMML can react with 2MGA and other organic compounds through oligomerization processes (e.g., Chan et al. (2010); Nguyen et al. (2015); Zhang et al. (2011, 2012)). Some of these organic oligomerization reactions bring into the particle phase additional organic compounds (e.g., organic acids) that ordinarily would exist primarily in the gas phase (Figure 2). The details of these particle-phase reactions are further discussed in Section 5.2.

In general, there is much greater variability in the SOA mass yields measured from the 2MGA pathway than the LV pathway. The additional variability is only partially explained by the initial seed surface area (Figure 7). Because the SOA yield is larger for experiments in which less methacrolein is oxidized (Figures 6 and 7b), potentially, the extent of methacrolein oxidization contributes to this variability. The kinetic model suggests that formation of gas-phase HMML is similar for all of the experiments (Figure S4), but potentially slight variations in the $NO_2/NO$ ratio and/or OH particularly near the end of each experiment are not well captured by the model. The kinetic model used here only simulates gas-phase oxidation. Chemistry occurring on surfaces such as the chamber walls or in the particle phase may be especially important for capturing the variability in the 2MGA pathway experiments. Considering that the 2MGA pathway experiments are very susceptible to small differences in chamber conditions, regional and global models should parameterize SOA formation from the 2MGA pathway through gas-phase formation of HMML and subsequent particle-phase reactions.

# 5  Discussion

The gas-phase compounds measured by the CIMS (Section 5.1) and aerosol composition measured by the AMS (Section 5.2) provide important insight into isoprene SOA chemical composition formed from both the LV and 2MGA pathways. Additionally, comparison of the AMS and DMA results lends insight into possible biases in the AMS measurements of organic aerosol in Section 5.3. The SOA yields measured in this study are compared with past measurements in Section 5.4 and the atmospheric contribution of the LV versus 2MGA pathways toward SOA formation from isoprene OH-initiated oxidation under high-$NO_x$ conditions is estimated in Section 5.5.

## 5.1 Specific Low-Volatility Nitrates and Dinitrates Detected in the Gas-Phase

Numerous nitrates and dinitrates are detected in the gas phase by the $CF_3O^-$ CIMS (i.e., compounds highlighted in blue boxes in Figure 1). Many of these nitrates have been identified in previous studies (e.g., Lee et al. (2014)). Yields for the low-volatility later-generation nitrates are either highly uncertain or unknown. Quantification is difficult for these low-volatility compounds due to high losses to sampling lines or chamber walls and lack of available standards. One study, Lee et al. (2014), was able to quantify the yield of dinitrates from the first-generation isoprene hydroxy nitrate standards. Assuming a sensitivity similar to the isoprene hydroxy nitrate standards, Lee et al. (2014) measured a dinitrate yield of 0.03-0.04 from OH-initiated oxidation of the $\delta$-1-hydroxy,4-nitrate isomer.

Although most past studies have focused on dihydroxy dinitrates as the main contributor to isoprene high-NO SOA, other low-volatility nitrates are likely also important. In Figure 8, the CIMS signals for the other low-volatility nitrates are comparable or larger than the dihydroxy dinitrate signal. The relative sensitivities for these compounds are unknown, but these results suggest that detection and quantification of all low-volatility dinitrates and nitrates is important. The peroxy radical formed from OH-initiated oxidation of an isoprene hydroxy nitrate can undergo a 1,5 or 1,6 $\alpha$-hydroxy H-shift to form a number of low-volatility nitrates that would occur in the ambient atmosphere (Wennberg et al., 2018). The NO concentrations are too high in these experiments for such shifts to occur. However, similarly, certain isomers of the alkoxy radical, formed from OH-initiated oxidation of a isoprene hydroxy nitrate, can undergo a 1,5 $\alpha$-hydroxy H-shift to form a dihydroxy carbonyl nitrate detected by the CIMS at m/z (-) 264 (Figure 1 and 8). Additionally, various low-volatility nitrates in the gas phase are detected, which are potentially oxidation products from the $\delta$-isoprene hydroxy alkoxy radical as depicted in Figure 1.

Many multi-functional isoprene derived organic nitrates have been detected in ambient aerosol (Lee et al., 2016). Although these low-volatility nitrates and dinitrates have low molar yields from isoprene OH-initiated oxidation, their mass is substantially larger than isoprene and so their contribution to the isoprene SOA mass yield is significant. The nitrate yield from straight chain hydrocarbons is reasonably well understood, but few experimental measurements of the nitrate yield from highly oxidized compounds exist (Orlando and Tyndall, 2012; Wennberg et al., 2018). Further measurements of the yield of these low-volatility nitrates and dinitrates in the gas phase will be crucial for better understanding isoprene SOA formation under high-NO conditions.

## 5.2 Aerosol Composition of High-NO Isoprene SOA

Pieber et al. (2016) determined that inorganic aerosol such as ammonium nitrate or ammonium sulfate causes an interference on the AMS for the $CO_2^+$ ion signal. Although this interference is small for ammonium sulfate aerosol (<1%, (Pieber et al., 2016)), a correction may be needed for experiments with high initial seed aerosol loadings. Here organic signals from the AMS rise when ammonium sulfate seed is injected into the chamber. We expect this is due to the same interferences described in Pieber et al. (2016) and not due to contamination in ammonium sulfate solution or atomization technique. The background organic signal caused by the ammonium sulfate is subtracted from the overall results to produce Figures 9, 10, and 11.

The AMS spectra from the LV pathway confirm that SOA formed from the LV pathway is not dominated by 2-MGA and its oligomers (cyan and red bars in Figure 9 and S11). This is an important confirmation that isoprene SOA formed from the 2MGA and LV pathways are distinct. A small yield of isoprene epoxydiol (IEPOX) is produced from OH-initiated oxidation of isoprene hydroxy nitrates (Jacobs et al., 2014) and IEPOX SOA can be formed when particle liquid water is present (Nguyen et al., 2014a). NO levels remained high (> 100 ppb) throughout all LV pathway experiments (Figure S3), ensuring that the $RO_2$ fate in these experiments was always $RO_2$ + NO. AMS fragments associated with IEPOX, which were identified by Lin et al. (2012), are slightly enhanced under humid conditions in the LV pathway experiments (Figure 9). Some examples of organonitrate fragments ($C_xH_yNO_z^+$) are highlighted in Figure 9. Some of these organonitrate fragments are enhanced under humid conditions (e.g., $CH_3NO^+$). In general, the AMS spectra are similar between all LV pathway conditions (i.e. varied humidity-Figure 9 and varied temperature - Figure S11).

Prominent peaks in the AMS spectra from the 2MGA pathway clearly indicate that under dry conditions aerosol is comprised of various oligomerization products as mechanistically summarized in Figure 2. These oligomerization processes include 2-MGA oligomerization with HMML (cyan bars in Figure 10a) and, possibly, esterification of 2-MGA with carboxylic acids including formic, acetic, and pyruvic acids (red bars in Figure 10a), which yield products that have been detected in numerous studies (Chan et al., 2010; Zhang et al., 2011, 2012). Based on the AMS spectra, 2-MGA oligomerization appears to be more dominant without the presence of ammonium sulfate seed aerosol (Figure S13). Varying temperature from 13-32°C does not appear to substantially change the extent of 2-MGA oligomerization (Figure S14).

Past studies have determined that HMML reaction with 2-MGA to form oligomers decreases under humid conditions while HMML ring-opening reactions with water and inorganic ions to form organic nitrates and organic sulfates increase (Zhang et al., 2011, 2012; Nguyen et al., 2015). Consistent with these past studies, the 2-MGA oligomer fragments on the AMS (cyan and red) are no longer prominent signals for all humid experiments (Figure 10 and Figure S12). 2-MGA oligomer fragments are not substantially different at 47%, 67%, or 81% RH, suggesting that the HMML oligomerization processes are impeded as soon as aerosol particles become deliquesced. Because isoprene is mostly emitted in regions with relatively high humidity, in the ambient atmosphere, HMML will more likely react with water and inorganic ions than undergo the various organic oligomerization reactions summarized in Figure 2.

### 5.3 Comparison of AMS and DMA Results

Based on the DMA measurements when assuming the same density, the SOA mass produced from the 2MGA pathway experiments is ∼2 times higher in magnitude than that from the LV pathway experiments (Figures 4 and 6). However, the AMS results (Figure 11) suggest that the SOA mass produced from the 2MGA pathway experiments is ≥8 times larger than that from the LV pathway experiments. This implies that the collection efficiency (CE) and/or the ionization efficiency on the AMS is quite different between these two regimes. Because the AMS is significantly more sensitive to aerosol formed from the 2MGA pathway, and not to SOA formed from the LV pathway, even ambient organic aerosol measurements have the potential to be impacted. Understanding whether the AMS is systematically underestimating organic aerosol from organic nitrates and dinitrates in general, or if this is only relevant to the isoprene system is crucial as the AMS is used throughout the world to

quantify organic aerosol. Moreover, ambient measurements over the isoprene-rich Southeastern United States of particulate organic nitrates measured by the AMS are a factor of $\sim$5 lower than that measured by the thermal dissociation laser-induced fluorescence instrument (TD-LIF) (Lee et al., 2016). The relative CE differences between the LV and 2MGA pathways in this study and these field campaign results suggest further AMS calibration of organic nitrates is necessary.

In this work, a CE of 0.5 is assumed for both regimes consistent with past work (Nguyen et al., 2014b). The exact CE is not relevant as no mass yields are reported here from the AMS. Docherty et al. (2013) determined that the CE could be estimated based on the $f_{44}/f_{57}$ ratio. The $f_{44}/f_{57}$ ratio for all experiments (2MGA and LV) is $\geq 6$, which is where the CE vs $f_{44}/f_{57}$ curve plateaus at 0.2. Thus, the CE vs $f_{44}/f_{57}$ relationship developed by Docherty et al. (2013) is not able to explain the large difference in AMS sensitivity between aerosol formed from the LV and 2MGA pathways.

## 5.4   Comparison to Previously Reported SOA Yields

SOA mass yields reported from past environmental chamber studies of OH-initiated oxidation of isoprene under high-$NO_x$ conditions vary over the range of 0.001-0.41 (Bregonzio-Rozier et al., 2015; Clark et al., 2016), suggesting isoprene SOA yields are highly dependent on chamber conditions (Carlton et al., 2009). In Table 2, past reported SOA mass yields are summarized along with the chamber conditions for both isoprene and methacrolein OH-initiated oxidation under high-$NO_x$

conditions. Only experiments that explicitly measure an SOA mass yield are listed in Table 2. Overall our results suggest that the initial seed surface area has the greatest impact on SOA yield. Unfortunately, the initial seed surface area was not commonly reported in past studies. The closest metric is aerosol volume, which can roughly be used to understand differences.

As shown in Table 2, the range for isoprene SOA yields under high-$NO_x$ conditions even from the two most recent studies at comparable temperatures spans over an order of magnitude (0.004 at $\sim$21$^\circ$C for Bregonzio-Rozier et al. (2015) and 0.1 at 27$^\circ$C

for Clark et al. (2016)). Our results are most consistent with those of Clark et al. (2016). As shown in Table 2, a variety of $NO_x$ regimes (i.e., non-consistent $NO_2/NO$ ratios) are all labeled as high-$NO_x$ in these past studies. Each study likely produces SOA in varying degrees from the LV and 2MGA pathways, which greatly complicates direct comparison between these past studies. By varying a large number of conditions and completely separating SOA production between the 2MGA and LV pathways, our results lend insight into the variation in these past experiments.

Many of the past SOA yield measurements were performed with no seed aerosol. Consistent with past results, when no seed aerosol was injected into the chamber (experiments D1 and M1), the SOA mass yield for the LV pathway (0 from isoprene) and 2MGA pathway (0.1 from methacrolein) were quite low. Past experiments performed with no seed aerosol were only measuring SOA from the 2MGA pathway, which is highly dependent on the $NO_2/NO$ ratio (Chan et al., 2010), which varied greatly between these past studies. Clark et al. (2016), who measured high SOA yields (0.1 at 27$^\circ$C) in unseeded experiments

is the exception. Possibly, the larger chamber volume (90 m$^3$) used by Clark et al. (2016) compared to most studies listed in Table 2 reduced vapor wall losses and contributed to the enhanced SOA yield. However, other chamber characteristics might also be important because Zhang et al. (2011) measured quite low isoprene SOA yields (0.007-0.03) using a chamber larger than the one used in the Clark et al. (2016) study.

While the zero or low seed aerosol loading experiments in this study generally compare well with the past, SOA yields measured here using higher initial seed surface areas are substantially greater than most studies, especially for the LV pathway. The SOA yield from the LV pathway is $\sim$0.15 in this study, while past isoprene SOA yields are largely $\leq$ 0.07 with the exception of studies optimizing for high $RO_2 + NO_2$ reactions (Chan et al., 2010) or mixed regimes - $RO_2 + HO_2/NO$ (Xu et al., 2014). The SOA yield from the LV pathway in this work is even larger than the SOA yield from Clark et al. (2016) (0.1 at 27$^\circ$C), which includes SOA from both the LV and 2MGA pathways. Possibly the larger chamber volume used by Clark et al. (2016) reduces vapor wall losses, but not to the extent that enhanced seed surface area does in this work. The higher yields measured in this study are not unexpected given that recent publications have recognized the importance of using high initial seed surface areas when measuring SOA yields to reduce the impact of vapor wall deposition (e.g., Zhang et al. (2014), Ehn et al. (2014)). The methacrolein SOA yields measured in this study from the 2MGA pathway are comparable to those measured by Chan et al. (2010), but larger than those measured by Bregonzio-Rozier et al. (2015).

Bregonzio-Rozier et al. (2015) measured low isoprene (0.001-0.01) and methacrolein (0.005-0.042) SOA mass yields and proposed that these lower yields were due to using xenon arc lamps as a light source, which are more representative of natural sunlight than the UV lamps used here and in most other studies. Dommen et al. (2006) also used xenon arc lamps and reported low yields. However, both of these studies used chambers with moderate to low chamber volumes (27-4.2 m$^3$) unlike the chamber used by Clark et al. (2016) and low levels of initial seed aerosol (0-16 $\mu$m$^3$/cm$^3$) unlike this work, which could also cause this low bias. Additionally, the stainless steel chamber used by Bregonzio-Rozier et al. (2015) may have higher vapor wall losses than the Teflon chambers used in other studies. Further work is necessary to understand how vapor wall losses compare across different types of environmental chambers.

As discussed by Carlton et al. (2009), isoprene SOA forms mostly from oxidation of 2$^{nd}$ and later-generation products (e.g., Ng et al., 2006). Towards the end of the experiment, SOA continues to grows even when isoprene is no longer reacting (e.g., the characteristic hook in Figure 4). Differences in the level of oxidation of 2$^{nd}$ and later-generation products could also explain some of the discrepancies between our results and past results. The isoprene SOA mass yields from the LV pathway are particularly sensitive to the extent of oxidation. More studies measuring the gas-phase yields and formation processes of low-volatility nitrates and dinitrates will be critical for further understanding isoprene SOA.

Many of the previous studies listed in Table 2 report the VOC/NO ratio when comparing experiments. A more useful metric is understanding the $RO_2$ fate and $RO_2$ lifetime. Simply injecting NO and/or $NO_2$ and reporting the initial concentrations are not sufficient to confirm that SOA was dominantly produced from the $RO_2 + NO$ channel or in the case of HMML formation from the $RO_2 + NO_2$ channel. For example, if NO decreases to zero before the end of the experiment, SOA has formed in a mixed regime; $RO_2 + NO$ reactions dominate in the beginning and $RO_2 + HO_2$ reactions dominate at the end. If large initial VOC loadings are used in the beginning of the experiment without comparable increases in NO, $RO_2 + RO_2$ reactions may become dominant.

Experiments here are specifically designed to test two different $RO_2$ fates, and the kinetic mechanism is used to confirm the fate of the $RO_2$. In the LV pathway experiments, high NO levels are maintained such that $NO_2/NO$ ratio remains < 1.5 throughout the entire experiment and $RO_2$ dominantly and consistently across the experiments reacts with NO. In the 2MGA

pathway experiments, high $NO_2$ levels are used such that the acyl radical derived from methacrolein dominantly and consistently across experiments reacts with $NO_2$. By controlling for the $RO_2$ fate, the effects of temperature, seed surface area, and relative humidity on SOA formation become easier to resolve. Design of future experiments should optimize and report the $RO_2$ fate for which the experiment was designed, in addition to key reaction parameters such as seed surface area, rather than

simply reporting an initial VOC/NO ratio.

## 5.5   Estimating the Atmospheric Contribution of the LV versus 2MGA Pathways

This work was not only designed to independently study SOA formation from the two high-$NO_x$ regimes (the 2MGA and LV pathways), but also to suggest alternative methods for parameterizing isoprene SOA under high-$NO_x$ conditions in regional and global models. Because obtaining constant $NO_2$/NO ratios similar to the ambient atmosphere is near impossible for a

chamber study (e.g., temporal variation in Figure S3), creating isoprene SOA parameterizations based on $NO_2$/NO ratio that realistically extrapolate to the ambient atmosphere is not realistic. Instead, this work highlights a potential alternative. Aerosol from the 2MGA pathway could be incorporated directly from gas-phase HMML formation and aerosol from the LV pathway could be included either from formation of surrogate compounds such as isoprene dihydroxy dinitrates or with a volatility basis set scheme. By treating the SOA from these two independent regimes separately, this study sets up the experimental basis for

such an approach.

    In this study, direct comparison of the results from the 2MGA and LV pathways is difficult due to the difference in the extent of oxidation between the two regimes caused by the use of different VOC precursors and the variation in OH levels (Table 1). Thus, the kinetic model is used here to estimate the contribution of each pathway to the total under consistent oxidant levels. A detailed global modeling study is needed to precisely capture the contribution of the LV versus the 2MGA pathways

toward SOA formation from isoprene OH-initiated oxidation under high-$NO_x$ conditions. However, in order to demonstrate the significance of the new isoprene SOA yield from the LV pathway measured in this work, we roughly approximate the contribution of each pathway under typical atmospheric conditions. We use the same kinetic mechanism described in Section 3, but hold the following constant: RH = 70%, T = 298K, $NO_2$ = 0.3 ppb, NO = 0.05 ppb, isoprene = 5 ppb, OH = 1.5 x $10^6$ molec $cm^{-3}$, CO = 135 ppb, $O_3$ = 37 ppb, and $HO_2$ = 25 ppt (Sanchez et al., 2018; Feiner et al., 2016; Pajunoja et al.,

2016). Then gas-phase HMML and the gas-phase dinitrate SOA precursors are simulated as done for the experimental results in Figure S4.

    To estimate the aerosol contribution from the LV pathway, we assume that SOA production from the LV pathway scales with the production of isoprene dihydroxy dinitrates. Organic aerosol concentrations are higher in chamber experiments than the ambient atmosphere. By using low levels of VOC precursors compared to previous studies, this study attempts to reduce

the organic aerosol concentrations to produce results more relevant to the ambient atmosphere. However, due to limitations in the DMA sensitivity, reducing the organic aerosol concentrations further to ambient levels is not possible. The ratio of the measured SOA yield (Figure 5) versus the simulated gas-phase dihydroxy dinitrate SOA precursor yield (Figure S4) is about 5. $F_P$ is decreased by a factor of 2 for the dihydroxy dinitrates when $C_{OA}$ is reduced from ~25 μg $cm^{-3}$ in the chamber to ~4 μg $cm^{-3}$ measured in the Southeast U.S. (Zhang et al., 2018). Thus, we multiply the dihydroxy dinitrate SOA precursors

by 2.5 and we convert to mass by multiplying by the molecular weight of dihydroxy dinitrate. MCM v3.3.1 assumes a nitrate yield of 0.087-0.104 from NO reacting with the peroxy radical derived from OH + isoprene hydroxy nitrate. Low-volatility nitrates such as dihydroxy hydroperoxy nitrates form when $HO_2$ reacts with the peroxy radical derived from OH + isoprene hydroxy nitrate. Such products would not form in the chamber conditions used in this work where NO levels remained above

100 ppb, but would form in the ambient atmosphere. Considering these low-volatility species from mixed chemical regimes would further increase the SOA mass generated from the LV pathway.

For the 2MGA pathway, we convert to mass by multiplying gas-phase HMML by the molecular weight of 2-MGA (120 g/mol), 2-MGA-nitrate (165 g/mol), and 2-MGA-sulfate (200 g/mol), which are the expected condensed-phase products under the high humidity levels in the atmosphere. Laboratory studies confirm that 2-MGA forms under humid conditions and some

of the 2-MGA partitions to the gas-phase as expected based on its volatility (Nguyen et al., 2015). For simplicity, we assume most of the HMML forms 2-MGA-nitrate and 2-MGA-sulfate, but acknowledge further experimental and modeling studies are needed to fully understand HMML/2-MGA aqueous phase chemistry.

Then based on the gas-phase SOA precursor distribution from the kinetic model and assumptions above, under typical atmospheric conditions the fraction of the total SOA mass from isoprene OH-initiated oxidation under high-$NO_x$ conditions is

~0.7 from the LV pathway and ~0.3 from the 2MGA pathway. This assumes that the dihydroxy dinitrates are valid surrogates for the isoprene SOA. Considering many multi-functional isoprene derived organic nitrates have been detected in ambient aerosol (Lee et al., 2016), all SOA precursors in Table S2 with $F_P$ > 0.05 at 26°C are combined and converted to mass. Extrapolating these to ambient organic aerosol concentrations is more difficult because these compounds are more likely to exist in the particle phase because of accretion reactions and not volatility. When these products are assumed to exist entirely in

the particle-phase and no factor is applied to correct for differences in organic aerosol concentration or for these products only representing about 1/3 of the isoprene SOA yield measured in this study (Figure S4), the LV pathway is estimated to contribute to ~0.6 of the SOA formed under high-$NO_x$ conditions.

Thus, based on the simple calculations summarized above, the LV pathway may produce moderately more SOA mass than the 2MGA pathway in the atmosphere and consequently deserves equal attention. The conditions chosen here represent

average atmospheric conditions around noon as measured during the SOAS field campaign, which occurred in the isoprene-rich Southeastern United States (Sanchez et al., 2018; Feiner et al., 2016; Pajunoja et al., 2016). A more complete assessment using global and regional modeling is needed to more definitively determine the fraction of SOA formed via the LV versus 2MGA pathways as location, time of day, season, ambient aerosol concentration and composition, etc. will all impact the amount of SOA formed from each pathway. Additional studies addressing organic nitrate hydrolysis and aerosol acidity are also necessary

to fully understand the relative impact of the two pathways on SOA formation. Additionally, the kinetic model used in this work only estimates gas-phase potential SOA precursors. Future analysis using a more complex model that explicitly simulates both the gas and particle phases would be useful for extrapolating the SOA yields measured here to the ambient atmosphere, which typically has lower organic aerosol concentrations than chamber experiments. This would need to be combined with additional analysis of the chemical constituents in the particle phase. From past work (Kleindienst et al., 2009; Xu et al., 2014; D'Ambro

et al., 2017) demonstrating that isoprene derived SOA under high-$NO_x$ conditions is lower in volatility than that derived under

low-NO$_x$ conditions and the C* values estimated in this work (Table S2), accretion reactions appear to be important even in the LV pathway experiments. The degree to which accretion reactions occur in the LV pathway experiments to form even lower volatility products is quite uncertain and will greatly impact future analysis on how best to extrapolate isoprene SOA yields measured in chambers to the ambient atmosphere.

## 6   Conclusions

SOA from OH-initiated isoprene oxidation under high-NO$_x$ conditions forms from two major pathways: 1) low-volatility nitrates and dinitrates (LV pathway) and 2) 2-methyl glyceric acid and its oligomers (2MGA pathway). These SOA production pathways respond differently to experimental conditions, so this work examines the SOA yields from these two pathways independently. Results suggest that low-volatility nitrates and dinitrates produce significantly more aerosol than previously thought with the isoprene SOA mass yield from the LV pathway ∼0.15. Sufficient initial seed aerosol is necessary to reduce the impact of vapor wall losses of low-volatility compounds and accurately measure the entire SOA mass yield. Even though previous studies have assumed that isoprene high-NO$_x$ SOA largely forms from 2-MGA and its oligomers (Chan et al., 2010; Zhang et al., 2011, 2012), results from this study confirm low-volatility compounds are also important for isoprene SOA formed under high-NO$_x$ conditions. The fate of isoprene's RO$_2$ radicals and the environmental conditions will determine which pathways are active in the atmosphere at a certain time and location.

Under dry conditions, substantial amounts of SOA form from HMML reaction with 2MGA to produce oligomers. The AMS results confirm that under humid conditions, these low-volatility oligomers are diminished in favor of higher-volatility monomer formation (and potentially subsequent volatilization of 2-MGA) to reduce the SOA mass. Thus, under atmospherically relevant humid conditions, aerosol formed from the 2MGA pathway is limited to HMML reaction with water or inorganic ions such as nitrate and sulfate. The importance of SOA from the 2MGA pathway will also depend on the NO$_2$/NO ratio, while SOA formed from the LV pathway will be important under all NO$_2$/NO ratios. Under typical atmospheric conditions (RH = 70%, T = 298K, NO$_2$/NO = 6, NO = 0.05 ppb, isoprene = 5 ppb, and OH = 1.5 x 10$^6$ molec cm$^{-3}$), we now estimate based on the simple assumptions discussed in Section 5.5 that the LV pathway produces moderately more SOA mass than the 2MGA pathway due to the high isoprene SOA yield from the LV pathway measured in this work.

Given the high isoprene SOA mass yield from the LV pathway (∼0.15) measured here, low-volatility compounds are as important as 2MGA-based compounds for isoprene SOA formed under high-NO$_x$ conditions. Thus, further studies investigating the formation rates and yields of these low-volatility compounds are needed. Consistent with past work (e.g., Lee et al. (2014)), a number of low-volatility nitrates and dinitrates, which are likely important precursors for SOA formed from the LV pathway, were detected in the gas phase by the CF$_3$O$^-$ CIMS (Section 5.1). These low-volatility compounds are likely derived from OH-initiated oxidation of the first-generation isoprene hydroxy nitrates. Synthetic pathways toward standards of many of the isoprene hydroxy nitrates exist (Lee et al., 2014; Teng et al., 2017). Now that this study has confirmed that low-volatility products contribute significantly to isoprene SOA, measuring SOA mass yields under varying RO$_2$ fate using these isoprene hydroxy nitrate standards as the initial precursor instead of isoprene could be particularly valuable for decreasing the uncer-

tainty in isoprene SOA yields. Additionally, an improved mechanistic understanding of isoprene SOA is needed. This would include an improved understanding of gas-phase reactions including measurements of highly functionalized peroxy radical isomerization rate constants, quantification of nitrate and hydroperoxide yields from highly functionalized $RO_2$ radicals reacting with NO or $HO_2$, respectively, and additional constraints on possible particle-phase accretion reactions leading to lower volatility products (e.g., hemiacetal formation).

There are some limitations for how results from this study should be interpreted. In the atmosphere, the $RO_2$ lifetime is longer than that in chamber experiments from this study and most past studies measuring SOA yields. Due to limitations in the sensitivity of the DMA and high NO levels needed to control the $RO_2$ fate, performing SOA yield chamber experiments at conditions that favor a long $RO_2$ lifetime is difficult. At longer $RO_2$ lifetimes, the hydroxy nitrate isomer distribution shifts toward a higher percentage of β-isomers over δ-isomers (Peeters et al., 2014; Teng et al., 2017). Additionally, $NO_x$ emissions are decreasing across many regions of the world due to improvements in emissions controls creating mixed regimes in the ambient atmosphere where a later-generation gas-phase product could form from $RO_2$ + NO reaction during the 1[st]-generation and $RO_2$ + $HO_2$ during the 2[nd]-generation. Field measurements confirm the presence of such products. For example, Xiong et al. (2015) discuss the presence of dihydroxy hydroperoxy nitrates detected in the particle phase by Lee et al. (2016) during SOAS, a field campaign that occurred during the summer in the Southeastern United States. Dihydroxy hydroperoxy nitrates likely form when hydroxy nitrates, produced from the $RO_2$ + NO pathway, react with OH and $O_2$ to form a peroxy radical that then reacts with $HO_2$. Additionally, because isoprene SOA from the LV pathway only forms once later-generation products get oxidized, the extent of oxidation is important, but also difficult to compare across different studies.

Comparisons of the DMA and AMS results imply that the collection and/or ionization efficiency on the AMS for SOA formed from the LV pathway is significantly lower than that formed from the 2MGA pathway. This could have important consequences on the interpretation of ambient organic aerosol measured by the AMS. Further work calibrating organic hydroxy nitrates on the AMS is needed to better understand why the organic fraction analysis varied so significantly between the two pathways.

Results from this work combined with past work provide further insight into how isoprene SOA should be parameterized in global and regional atmospheric chemistry models. Under humid conditions, SOA formation from the 2MGA pathway is produced mostly from HMML ring opening reactions to form monomer compounds 2-MGA, 2-MGA-nitrate, and 2-MGA-sulfate, which simplifies the parameterization of SOA from the 2MGA pathway as the organic oligomerization reactions can be ignored. The particle's liquid water and pH will be important to consider, as these metrics shift the equilibrium of 2MGA and its carboxylate and change the hydrolysis rates for the 2-MGA-nitrate and 2-MGA-sulfate. The gas-phase kinetics for MPAN formation and reaction with OH to form HMML have been reasonably well studied (e.g., Orlando et al. (1999, 2002); Nguyen et al. (2015)). HMML formation and generation of SOA in the atmosphere would be best incorporated into models by directly forming SOA through the MPAN + OH reaction. This would best parameterize the effects of temperature and $NO_2$/NO ratio on MPAN formation and also the influence of OH on HMML formation. This study confirms the need to perform experiments with adequate seed aerosol to limit vapor wall deposition processes when measuring SOA yields from the LV pathway. When regional chemical transport models use SOA yields that account for vapor wall deposition, there are

differences in the contribution of isoprene to the total SOA budget and improvements in the agreement between simulated and observed total SOA and diurnal variability (Cappa et al., 2016). Incorporating the isoprene SOA yields from the LV pathway measured in this work into models will further improve the accuracy of simulated isoprene SOA. Moreover, the results from this study along with future experiments studying the formation of low-volatility nitrates and dinitrates on a mechanistic basis

will be important for incorporating more explicit SOA formation into global models as has recently been done (e.g., Marais et al. (2016); Stadtler et al. (2018)), thus replacing previous parameterizations that typically were based on a single chamber condition (e.g.,Henze and Seinfeld (2006); Henze et al. (2008); Heald et al. (2008)).

*Data availability.*  We welcome future collaboration with those who wish to use this data set for additional modeling purposes (e.g., creating volatility basis set parameters for global/regional models or for evaluating the results with a more complex box-model that includes aerosol

chemistry). Please contact Rebecca Schwantes (rschwant@ucar.edu).

*Author contributions.*  RHS designed the experiments. RHS and SMC performed the experiments. RHS analyzed the data with help from SMC, KHB, TBN, JHS, and YH. RHS did the kinetic modeling. YH, HM, WK, and RCF assisted RHS with DMA operation and data analysis. RHS wrote the manuscript with assistance from KHB, TBN, JHS, SMC, and YH.

*Competing interests.*  The authors declare that they have no conflict of interest.

*Acknowledgements.*  This work was supported by the National Science Foundation grant AGS-1523500. Sophia M. Charan is supported by the National Science Foundation Graduate Research Fellowship under grant 1745301. We thank Dennis Fitz for assistance with maintenance and data analysis of the luminol $NO_2$/acyl peroxynitrate analyzer. The National Center for Atmospheric Research is sponsored by the National Science Foundation.

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

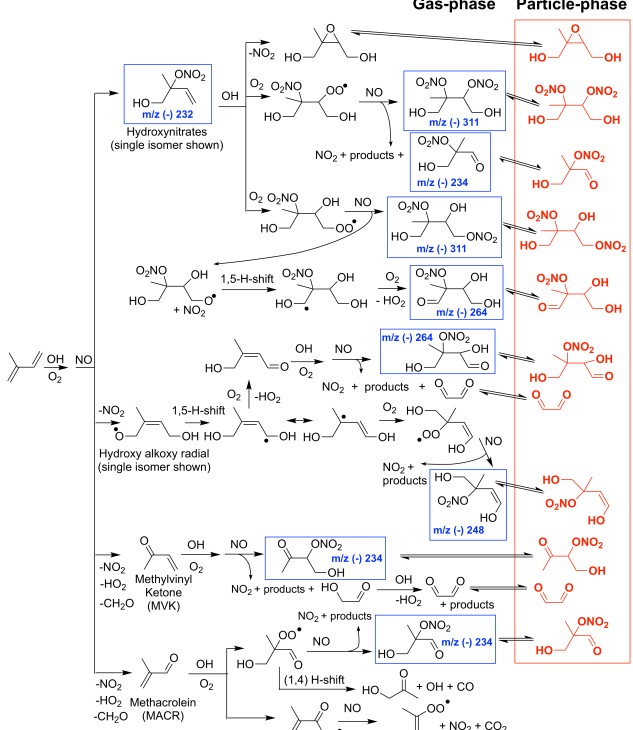

**Figure 1.** Simplified chemical mechanism of isoprene OH-initiated oxidation under high-NO conditions, largely based on schemes in Wennberg et al. (2018), emphasizing SOA generated from the LV pathway, which includes low-volatility organic nitrates and dinitrates in red. Compounds detected in the gas phase by the chemical ionization mass spectrometer (CIMS) are highlighted with a blue square.

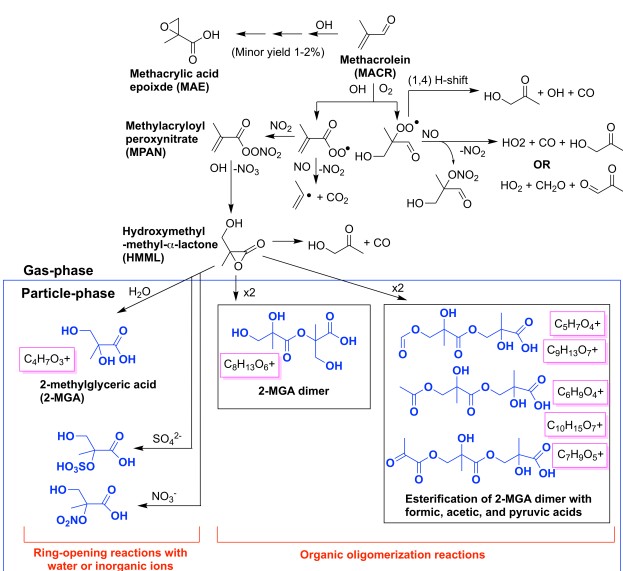

**Figure 2.** Simplified chemical mechanism of methacrolein OH-initiated oxidation under high-NO$_2$ conditions, largely based on schemes in Wennberg et al. (2018), emphasizing SOA generated from the 2MGA pathway including 2-methylglyceric acid (2MGA) and its oligomers in blue. Aerosol mass spectrometer (AMS) fragments likely corresponding to each compound are boxed in magenta.

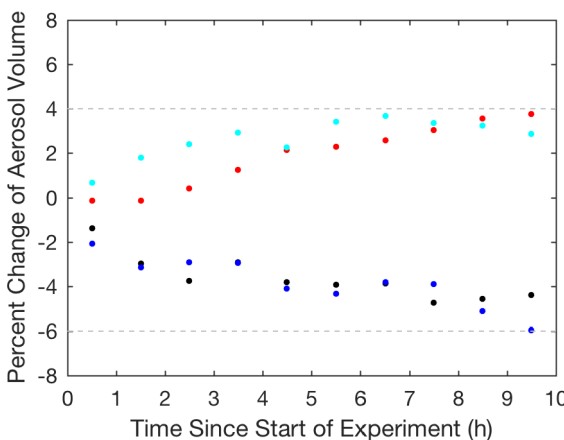

**Figure 3.** Percent change between the corrected aerosol volume over 10 h and the corrected aerosol volume at the start of photooxidation (60 min averages) for the following particle wall deposition control experiments: C1 (V = 37 $\mu m^3 cm^{-3}$, ●), C2 (V = 109 $\mu m^3 cm^{-3}$, ●), C3 (V = 183 $\mu m^3 cm^{-3}$, ●), and C4 (V = 375 $\mu m^3 cm^{-3}$, ●), respectively where V is the initial corrected particle volume.

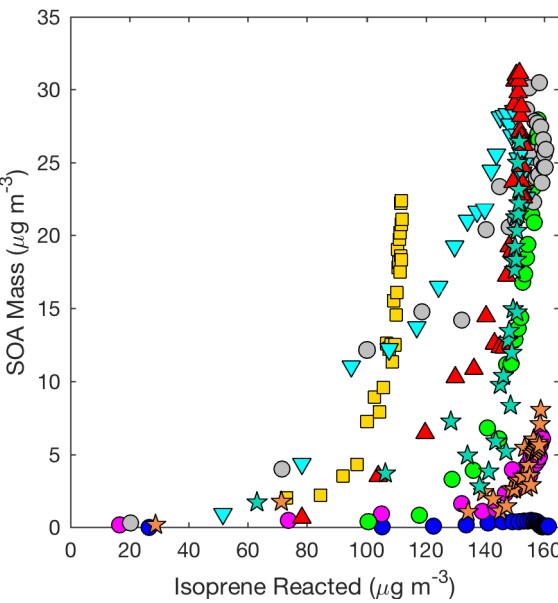

**Figure 4.** SOA mass yield (20 min averages) as measured by the DMA assuming a density of 1.4 g cm$^{-3}$ for all LV pathway experiments: seed surface area - D1 (SA = 0 μm$^2$cm$^{-3}$, ●), D2 (SA = 1170 μm$^2$ cm$^{-3}$, ●), D3 (SA = 3420 μm$^2$ cm$^{-3}$, ●), & D4 (SA = 5770 μm$^2$cm$^{-3}$, ●), temperature - D5 (13 °C, ▼) & D6 (32 °C, ▲), isoprene loading - D7 (initial isoprene 110 μg m$^{-3}$, ◆), and new chamber with less wall charging - D10 (SA = 1580 μm$^2$cm$^{-3}$, ★) & D11 (SA = 4770 μm$^2$cm$^{-3}$, ★)

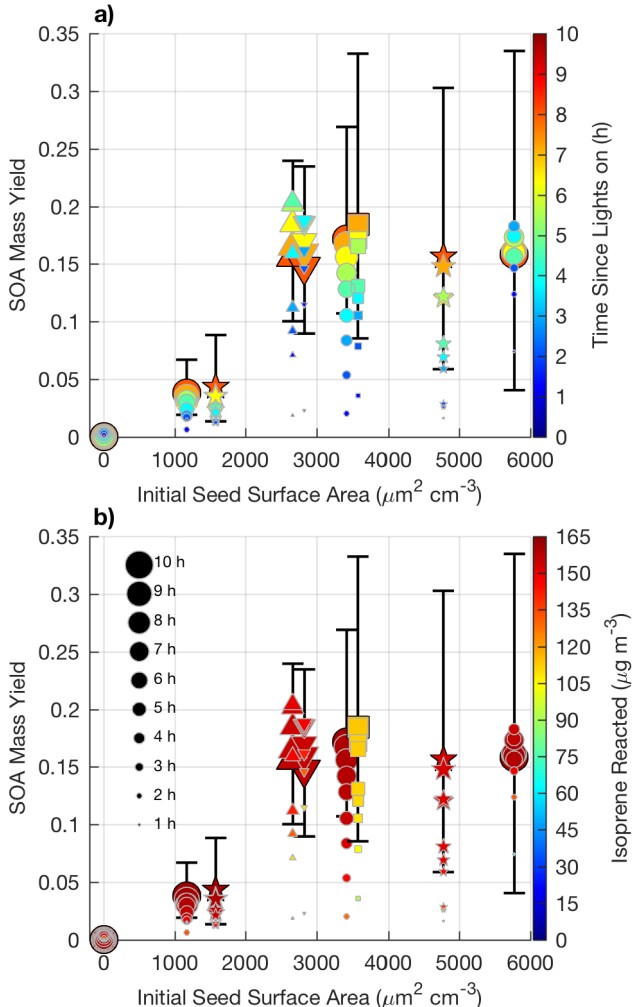

**Figure 5.** SOA mass yield (60 min averages) as a function of initial seed surface area for all LV pathway experiments. Colors represent time since lights on in panel a and extent of isoprene reacted in panel b. Marker size represents time since lights on. Uncertainty is shown in black lines as described in Section 4.1. Marker types indicate: 25-26°C (●), 13°C (▼), 32°C (▲), lower loadings of isoprene (■), and new chamber with less wall charging (★).

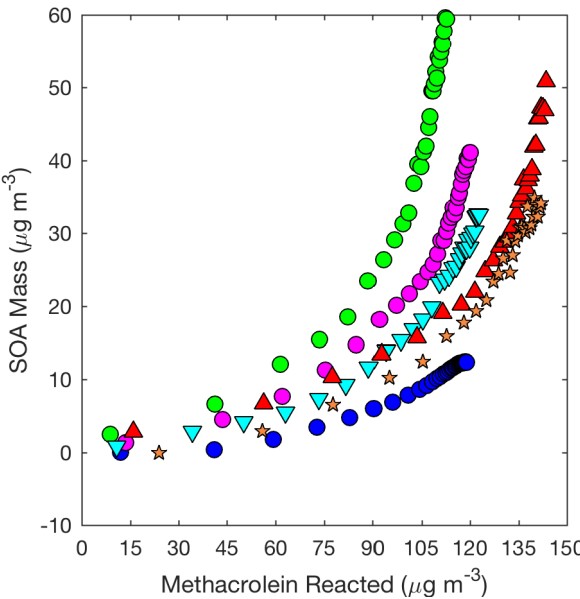

**Figure 6.** SOA mass yield (20 min averages) as measured by the DMA assuming a density of 1.4 µg cm$^{-3}$ for 2MGA pathway experiments: seed surface area - M1 (SA = 0 µm$^2$cm$^{-3}$, ●), M2 (SA = 1640 µm$^2$cm$^{-3}$, ●), & M3 (SA = 2260 µm$^2$cm$^{-3}$, ●), temperature - M5 (13 °C, ▽) & M6 (32 °C, ▲), and new chamber with less wall charging - M9 (SA = 1910 µm$^2$cm$^{-3}$, ★).

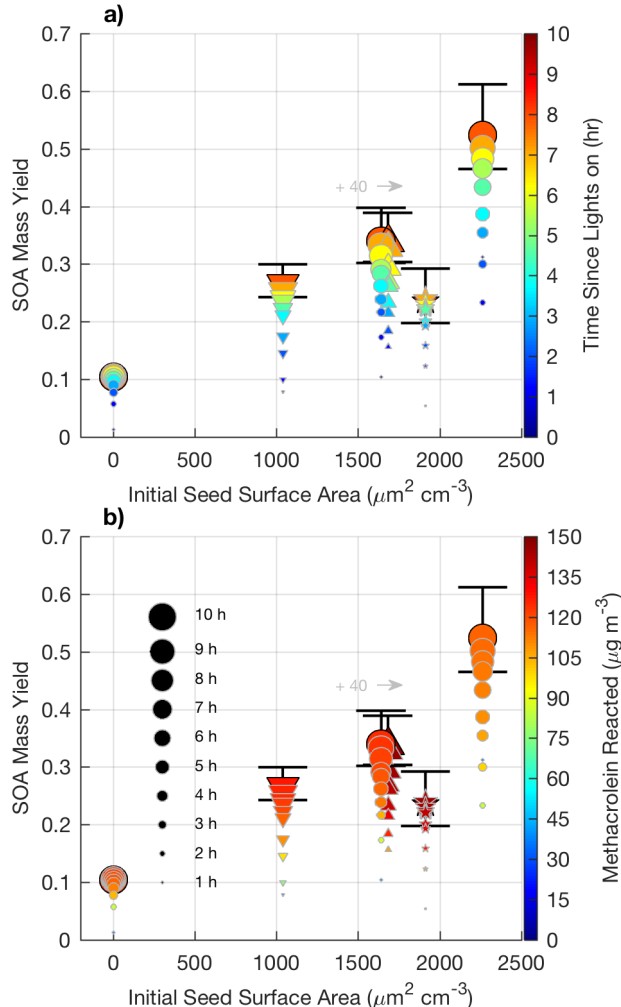

**Figure 7.** SOA mass yield (60 min averages) versus initial seed surface area for all 2MGA pathway experiments. Colors represent time since lights on (panel a) and extent of methacrolein reacted (panel b). Marker size represents time since lights on. Uncertainty is shown in black lines and described in Section 4.1. Markers represent: 25-26°C (●), 13°C (▼), 32°C (▲), and new chamber with less wall charging (★). Two experiments were performed at nearly the same seed surface area. To enhance viewing, experiment M6 (32°C, △) is shifted to the right by 40 μm$^2$ cm$^{-3}$.

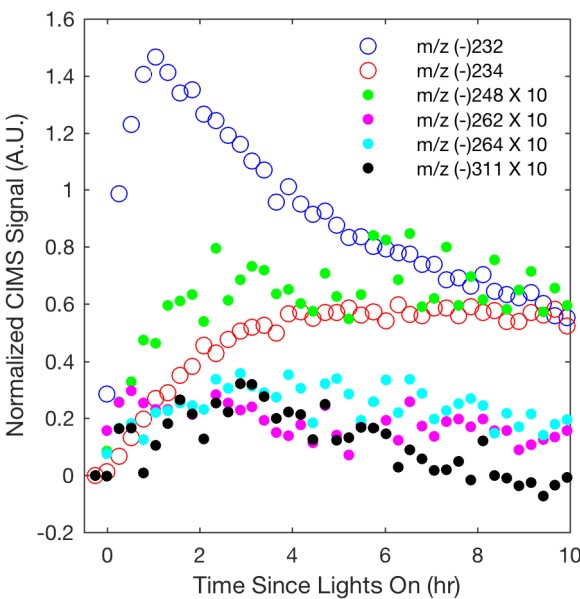

**Figure 8.** Normalized CIMS signal for known nitrates: $C_5$ hydroxy nitrate (m/z (-) 232 ∘), methyl vinyl ketone/methacrolein nitrate (m/z (-) 234 ∘), & $C_5$ dihydroxy dinitrate (m/z (-) 311 ●) and unknown nitrates, which are postulated in Figure 1 as $C_5$ dihydroxy nitrate (m/z (-) 248 ●), unknown (m/z (-) 262 ●), & $C_5$ dihydroxy carbonyl nitrate (m/z (-) 264 ●). As indicated in the legend, signals represented by filled circles are multiplied by 10.

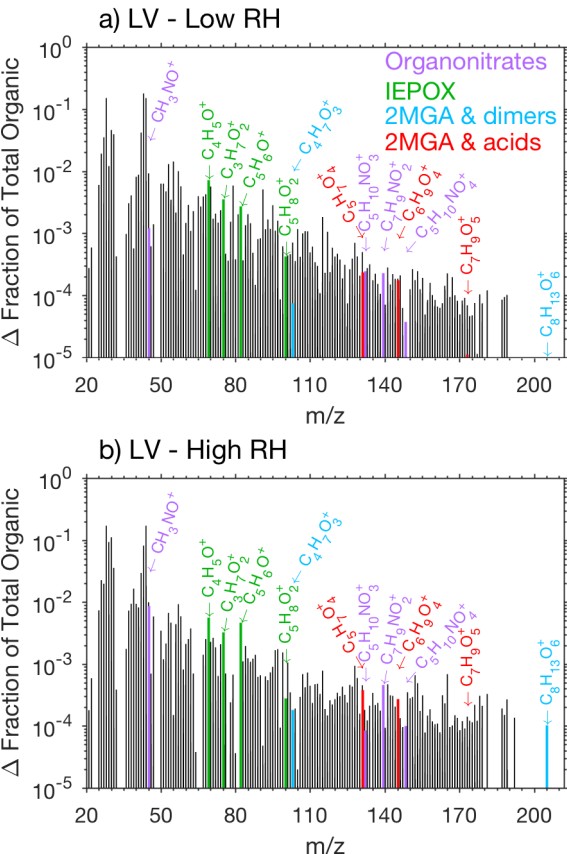

**Figure 9.** High resolution AMS mass spectra (averaged over 10h of photooxidation - the sulfate background) for experiment D3 (RH = 8%, panel a) and D9 (RH = 78%, panel b) in gray. Fragments are labeled as 2-MGA monomer/dimer (**cyan**), esterification of 2-MGA with acids (**red**), isoprene epoxydiol (IEPOX) tracers (**dark green**), and examples of organonitrate fragments - $C_xH_yNO_z$ (**purple**).

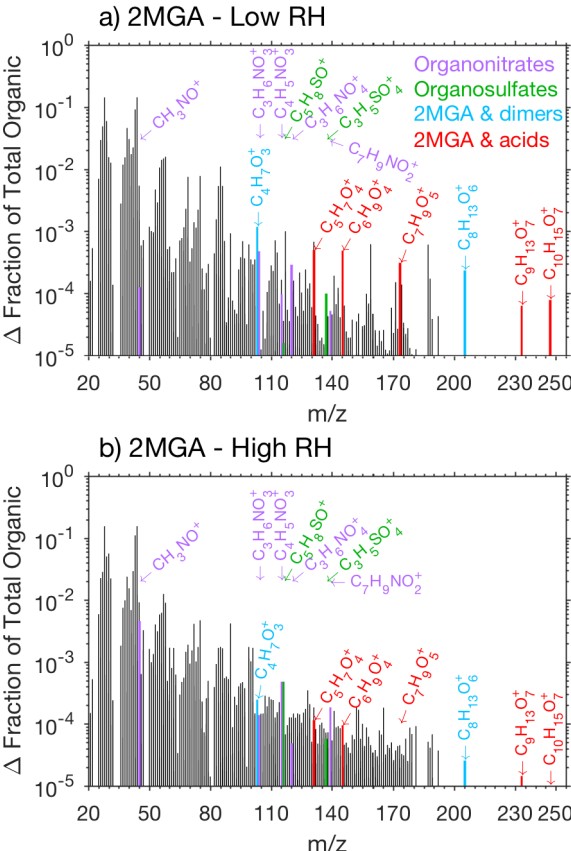

**Figure 10.** High resolution AMS mass spectra (averaged over 10h of photooxidation - the sulfate background) for experiment M2 (RH = 9%, panel a) and M8 (RH = 81%, panel b) in gray. Fragments are labeled as 2-MGA monomer/dimer (**cyan**), esterification of 2-MGA with acids (**red**), examples of organosulfate fragments (**dark green**), and examples of organonitrate fragments - $C_xH_yNO_z$ (**purple**).

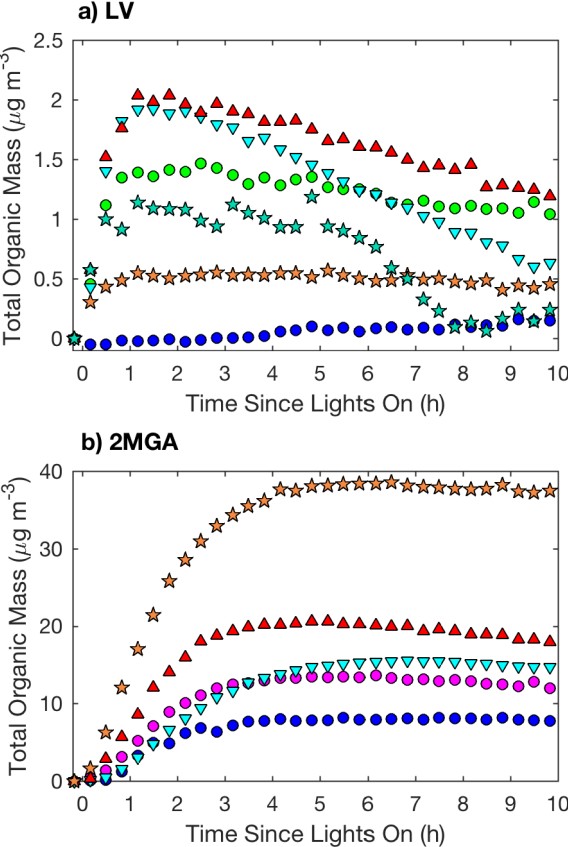

**Figure 11.** Total organic mass (20 minute averages) as measured by the AMS for LV pathway experiments (panel a): seed surface area - D1 (SA = 0 $\mu$m$^2$ cm$^{-3}$, ●) & D3 (SA = 3420 $\mu$m$^2$ cm$^{-3}$, ●), temperature - D5 (13 °C, ▼) & D6 (32 °C, ▲), and new chamber with less wall charging - D10 (SA = 1580 $\mu$m$^2$ cm$^{-3}$, ★) & D11 (SA = 4770 $\mu$m$^2$ cm$^{-3}$, ★) and 2MGA pathway experiments (panel b): seed surface area - M1 (SA = 0 $\mu$m$^2$ cm$^{-3}$, ●) & M2 (SA = 1640 $\mu$m$^2$ cm$^{-3}$, ●), temperature - M4 (13 °C, ▼) & M5 (32 °C, ▲), and new chamber with less wall charging - M9 (SA = 1910 $\mu$m$^2$ cm$^{-3}$, ★).

**Table 1.** Initial conditions and SOA yield for all experiments

| Expt # | [VOC]$_0$ (ppb) | [NO]$_0$ (ppb) | [NO$_2$]$_0$ (ppb) | [CH$_3$ONO]$_0$ (ppb) | [Aer Vol]$_0$ ($\mu$m$^3$ cm$^{-3}$) | [Aer SA]$_0$ ($\mu$m$^2$ cm$^{-3}$) | Avg T (°C) | Avg RH (%) | OH (molec cm$^{-3}$) | SOA Yield |
|---|---|---|---|---|---|---|---|---|---|---|
| **Dry Control Experiments** | | | | | | | | | | |
| C1 | NA | NA | NA | NA | 37 | 788 | 25.1 | 10.7 | NA | NA |
| C2 | NA | NA | NA | NA | 109 | 2130 | 25.2 | 8.3 | NA | NA |
| C3 | NA | NA | NA | NA | 183 | 3360 | 24.7 | 5.6 | NA | NA |
| C4 | NA | NA | NA | NA | 375 | 5390 | 25.5 | 7.3 | NA | NA |
| **Experiments optimized for LV pathway** (VOC precursor is isoprene) | | | | | | | | | | |
| D1 | 59 | 585 | 6 | 118 | 0 | 0 | 25.6 | 5.0 | 2.6E6 | 0 |
| D2 | 58 | 526 | 20 | 117 | 54 | 1170 | 26.4 | 5.6 | 2.5E6 | 0.04 |
| D3 | 57 | 519 | 17 | 117 | 183 | 3420 | 25.9 | 7.5 | 2.5E6 | 0.17 |
| D4 | 58 | 518 | 18 | 116 | 337 | 5770 | 26.4 | 7.9 | 2.4E6 | 0.16 |
| D5 | 55 | 506 | 20 | 117 | 159 | 2830 | 12.8 | 16.4 | 1.7E6 | 0.15 |
| D6 | 56 | 541 | 16 | 118 | 152 | 2660 | 32.4 | 5.9 | 2.7E6 | 0.16 |
| D7 | 40 | 527 | 18 | 117 | 197 | 3580 | 25.9 | 8.1 | 2.6E6 | 0.18 |
| D8 | 60 | 519 | 20 | 118 | 109 | 1790 | 25.5 | 44.7 | 2.3E6 | NA |
| D9 | 55 | 489 | 20 | 119 | 166 | 2750 | 25.6 | 78.1 | 2.5E6 | NA |
| D10 | 58 | 516 | 17 | 111 | 85 | 1580 | 25.8 | 5.1 | 2.2E6 | 0.04 |
| D11 | 56 | 490 | 17 | 115 | 264 | 4770 | 25.8 | 5.2 | 2.4E6 | 0.16 |
| **Experiments optimized for 2MGA pathway** (VOC precursor is methacrolein) | | | | | | | | | | |
| M1 | 49 | 14 | 376 | 234 | 0 | 0 | 25.8 | 6.3 | 4.3E6 | 0.10 |
| M2 | 48 | 15 | 365 | 235 | 82 | 1640 | 25.9 | 8.9 | 4.7E6 | 0.34 |
| M3 | 46 | 23 | 345 | 236 | 118 | 2260 | 25.1 | 6.8 | 4.7E6 | 0.52 |
| M4 | 50 | 17 | 356 | 235 | 50 | 1040 | 12.9 | 12.6 | 3.4E6 | 0.27 |
| M5 | 58 | 18 | 375 | 235 | 87 | 1740 | 31.8 | 4.5 | 5.1E6 | 0.34 |
| M6 | 52 | 12 | 334 | 235 | 104 | 1720 | 25.6 | 47.1 | 4.4E6 | NA |
| M7 | 53 | 14 | 339 | 233 | 134 | 2340 | 25.6 | 67.4 | 4.6E6 | NA |
| M8 | 56 | 18 | 352 | 236 | 141 | 2510 | 25.4 | 81.0 | 4.3E6 | NA |
| M9 | 57 | 29 | 298 | 229 | 95 | 1910 | 25.9 | 5.1 | 4.7E6 | 0.24 |

Acronyms are defined as follows: VOC = volatile organic compound, NO = nitric oxide, NO$_2$ = nitrogen dioxide, CH$_3$ONO = methyl nitrite, Temp. = temperature, and RH = relative humidity. OH (hydroxyl radical) is estimated from the VOC decay over the first 3 h of each experiment. The [Aer Vol]$_0$ is the particle wall loss corrected seed volume at the start of photooxidation, which is used to determine the uncertainty in the particle wall loss correction as explained in Section 4.1. The [Aer SA]$_0$ is the surface area of the seed aerosol at the start of photooxidation not corrected for particle wall loss, and is used to understand how the SOA yield changes depending on the surface area of the suspended particles (e.g., Figure 5). The SOA yield is the mass fraction after 10 h of photooxidation.

**Table 2.** Reported SOA mass yields and chamber conditions for isoprene and methacrolein OH-initiated oxidation under high-$NO_x$ conditions

| Study | CV (m$^3$) | Oxidant | [VOC]$_0$ (ppb) | [NO]$_0$ (ppb) | [NO$_2$]$_0$ (ppb) | Light Type | [AS]$_0$ (μm$^3$/cm$^3$) | Temp. (°C) | RH (%) | SOA Yield (fraction) |
|---|---|---|---|---|---|---|---|---|---|---|
| **Isoprene** | | | | | | | | | | |
| Edney (2005)[a] | 14.5 | NO$_x$ | 1610-1680 | ∼630 | 0 | UV | <0.6-24[b] | 29.7 | 30 | 0.002-0.028 |
| Kroll (2005) | 28 | HONO | 25-500 | 75-138 | 98-165 | UV | 10-25 | ∼20 | 40-50 | 0.009-0.03 |
| Dommen (2006) | 27 | NO$_x$ | 180-2500 | 0-700 | 40-806 | X | 0 | 20 | <2-84 | 0.002-0.053 |
| Kleindienst (2006)[a] | 14.5 | NO$_x$ | 1600 | 406-485 | 7-69 | UV | 0.1-27[b] | 25 | 30 | 0.003-0.018 [c] |
| Chan (2010) | 28 | HONO or CH$_3$ONO | 33-523 | 259-316 | 510-859 | UV | 11-19 | 20-22 | 9-11 | 0.031-0.074 |
| Chhabra (2010) | 28 | HONO | 81-286 [d] | 518-591 | 374-434 | UV | 11-14 | NR | <10 | 0.006-0.015 |
| Zhang (2011) | 137 | NO$_x$ | 400-790 | 138-253 | 1-9 | N | 10-30 | 281-303 | 15-88 | 0.007-0.03 |
| Nguyen (2011) | 5 | H$_2$O$_2$ | ∼250 | 600 | 100 | UV | 0 | 22-26 | <2-90 | ∼ 0.07 |
| Xu (2014) | 10.6 | H$_2$O$_2$ | 101-115 | 338-738 | 0 | UV | 0 | ∼25 | <5 | 0.015-0.085 |
| Bregonzio -Rozier (2015) | 4.2 | NO$_x$ or HONO | 439-846 | 14-143 | <1-79 | X | 0-16 | 16-24 | <5 | 0.001-0.01 |
| Clark (2016) | 90 | H$_2$O$_2$ | 250 | 500 | 0 | UV | 0 | 5-40 | dry | 0.1-0.41 |
| **Methacrolein** | | | | | | | | | | |
| Chan (2010) | 28 | HONO or CH$_3$ONO | 20-285 | 164-725 | 365-799 | UV | 11-16 | 20-22 | 9-11 | 0.019-0.392 |
| Bregonzio -Rozier (2015) | 4.2 | NO$_x$ or HONO | 396-927 | 19-123 | 4-100 | X | 0-15 | 19-24 | <5 | 0.005-0.042 |

CV = Chamber Volume. Acronyms are defined as follows: NR = not reported, UV = ultraviolet lights, N = natural, X = xenon arc lamps, and AS = ammonium sulfate seed aerosol volume. [a] Chamber was operated in dynamic mode (residence time = 6 h). [b] Ammonium sulfate was injected throughout the experiment to generate the lower limit of initial seed aerosol. SO$_2$ was added in some experiments to generate the upper limit of initial seed aerosol. [c] SOC is converted to SOA using factor (2.47) reported in Kleindienst et al. (2007). [d] VOC reacted was reported and tabulated instead of VOC initial.