# Peer review of "Low-volatility compounds contribute significantly to isoprene SOA under high-NOx conditions"

_Atmospheric Chemistry and Physics, 2018_

## Referee Comment (RC1) · Anonymous Referee #1 · 26 Jan 2019

This paper reports on the SOA yield of isoprene OH oxidation in a chamber study. The authors designed experiments to distinguish between different chemical pathways for a more systematic investigation than previous studies. In addition, experiments aimed for similar conditions during the time of the experiment. The authors conclude that the main difference previous results from chamber studies were due to the variability in the seed aerosol concentration.

The paper is well written und gives all required details. It is well suited for publication in ACP.

I have only few small comments:

[Figure]

p2 line 29: The authors may want to consider the publication of Peeters et al. PCCP 2014, if they mention new gas-phase chemistry of isoprene.

p3, Fig 1: I would suggest to add reference for the chemical scheme that is shown. What is the importance of the 1,5-H shift reaction that is shown in the scheme for conditions of high NO in these experiments?

p5 l6: I assume the specification of the Milli-Q water is meant to be 18 M Ohm.

p5 l18: I would suggest to add the mean diameter of the seed aerosol for information.

p7 l13: The author mention that in only few experiments NO2 was directly detected and in other NO2 was modelled. How was the model-measurement agreement of NO2 in the experiments, when NO2 was measured?

p8 l26: I would suggest to mention the parameters that were constrained by measurements in the model.

p12 l1-9: What does the unreasonable result of the correction of DMA data for wall loss mean for the uncertainty of yields determined for similar conditions in this work?

p14: Fig 5: The figure does not very clearly support the statement that SOA mass yield depends on the initial seed aerosol correction. There are essentially two values for the range from 1000 to 2000 and from 2500 to 6000 $\mu$m2/cm3. Could you please comment?

p25 l27-30: Does the statement that SOA from the LV pathway is moderately higher than from 2MGA takes into account the differences in the turnover of the OH oxidation of the precursors of the two pathways?

---

## Referee Comment (RC2) · Anonymous Referee #2 · 2 Feb 2019

Comments: This manuscript aimed to address contributions of isoprene SOA under high-NO conditions through forming the "low-volatility" organic nitrate compounds. The authors found that the isoprene SOA yields under tailored NOx conditions and substantial seed aerosol surface areas indeed are higher than previously estimated. The results are important and the paper is overall well written. But there are a few important issues that need to be addressed before published.

Major: 1. The authors concluded in the abstract that the LV pathway which produces organic nitrates and dinitrates likely contribute to isoprene SOA under high-NO conditions more substantially than previously thought under typical atmospheric conditions.

[Figure]

But some components of the typical atmospheric conditions were not thoroughly discussed. For example, how does hydrolysis of the organic nitrates affect this contribution? How would aerosol acidity affect the contribution from the 2MGA pathway? Would low particle surface areas in real atmosphere impede partitioning of the products that the authors discussed? In addition, as the authors have claimed themselves that the yields of these LV organic nitrates are highly uncertain, the MCM model assumes a yield of 10.4%. Estimation of LV pathway yield using this model is thus also highly uncertain. The results from this model that LV pathway yield of $\sim 0.15$ is rather hypothetic and I suggest leave it out of the abstract. Moreover, the partitioning behavior can be very different in the atmosphere compared to the chamber studies.

2. Validation is needed to explain why the authors only studied the 2MGA pathway using methacrolein as the initial VOC. I understand that the authors tried to tailor the conditions to separate the two pathways and measure SOA yield from each pathway. This is a really good idea. But then, a direct comparison between the two pathways is also needed, which is more straightforward and reasonable to start with isoprene and vary NO/NO2 between different experiments. For example, using the many analytical tools in the authors' lab, they can show that as NO/NO2 decreases, the gas-phase (CIMS) and particle-phase (AMS) organic nitrates and dinitrates decrease, while the HMML and 2MGA increase. Comparing with the isolated experiments, quantification of the two pathways is probably possible too. In this way, the same OH oxidation extent and seed particle surface area will be used and variations can be better controlled. Although the 2MGA pathway from isoprene and methacrolein oxidation is the same mechanistically, in direction isoprene photooxidation experiment, the 2MGA pathway products are formed in later generation and that may change the dynamics (e.g., OH availability at different times) in the SOA formation. Also, how signature SOA product ions (both CIMS and AMS) from the LV pathway and the 2MGA pathway in the same isoprene photooxidation experiments as NO/NO2 ratio varies is a more desired and direct way to present the results.

3. Page 12, line 22. It is very interesting that SOA yield does not vary too much as temperature changes between 13 and 32 C. The authors mention two reconciling effects: enhanced vapor wall loss and enhanced organic nitrate yields from RO2 + NO reaction under lower temperatures. Can the authors provide some quantitative constraints on these two effects? How different are the organic nitrate yields and wall loss rates under this range of temperatures? Another point to add, is that at lower temperature, more semi-volatile species may partition into the particle phase. I think this is a major effect. Previous studies using very similar temperature range and found a difference of 2-10 times in SOA yields (Clark et al., 2016 ES&T; Takekawa et al., 2003 Atmos. Environ.; Sheehan and Bowman, 2001 ES&T.). Some of these studies use isoprene, some use hydrocarbons that likely produce less volatile SOA than isoprene. The authors should think about addressing this question. Is it because the LV SOA have surprisingly low volatility that they are already mostly in the particle-phase at 30C? Can the authors estimate the vapor pressure of the nitrate and dinitrate compounds and see if this hypothesis make sense? Note that there are only two experiments at different temperatures and the lower temperature experiment started forming SOA much earlier than the higher temperature experiment. But they ended up forming similar SOA mass. From that perspective, there is a clear temperature influence. I hope the authors can address this question with more detailed discussion. The same case for the 2MGA pathway. The authors found little temperature influence on SOA mass yield. But again, that was based on only one high temperature and one low temperature experiment. The two experiments have very different seed particle surface areas too. The data provided are not sufficient to draw conclusions on temperature effects.

4. The authors use "low-volatility" to describe the organic nitrates. But they should also justify this by providing some estimates of the vapor pressures or C* of the molecules in Figure 1.

Minor: 1. Title and abstract: HMML is formed not only under high-NO, but also high-NO2 (MPAN is their precursor). It is thus inaccurate to term "high-NO" as the condition.

As the authors also point out in the text, NO2/NO ratio is an important indicator of the two pathways. From the experimental list, NO and NO2 are indeed adjusted to optimize different conditions. Thus, it seems more appropriate to use "high-NOx" rather than "high-NO" in this manuscript.

2. Page 3, line 2. The 2MGA pathway should refer to particle-phase products (2-MGA, its oligomers, and its organosulfates) from further oxidation of MPAN or uptake of HMML.

3. Page 5, line 27. Whether the NO2 signal can be interferences from nitrous acid or CH3ONO can be easily tested using the standards.

4. Page 12, line 14. It is unclear what is referred to as "the 2-MGA precursor" here. Is it MACR? MPAN? HMML? This information was obtained from the kinetic mechanism. I think the author should at least provide the simulations in the supporting information and mention it in the main text.

5. Page 12, line 26-27. Exceptions are when temperature was low and initial seed particle surface area was high.

6. Page 12, line 31. Based on the earlier description, the kinetic model combines MCM and some other gas-phase reactions that are listed in Table 2. So the kinetic model seems to be a pure gas-phase model. It is thus unclear when the authors mentioned that the kinetic model can predict the SOA mass yield. This sentence needs to be re-phrased to something like: "The kinetic model predicts that the formation of gas-phase dinitrates from experiment D7 is similar to the other experiments". Then the followed issue is whether this single species can represent the overall SOA in the studied condition. The authors may want to also compare other proposed LV-pathway products from the model.

7. Page 15, line 9. The authors may also want to consider that organosulfate formation in the presence of ammonium sulfate seed particles.

8. Page 15, line 15. HMML/reacted MACR = 0.25 does not mean the SOA yield from HMML uptake is 0.25. HMML is a C4H6O3 lactone. If you think about its vapor pressure, it is probably on the more volatile side of the SVOC range. It will have a large fraction in the gas phase.

9. Page 17, line 30 to Page 18, line 2. The description is unclear. A brief but better explanation (better than just saying "highly uncertain") is needed here to explain why the Lee et al. used upper limit yields 10 times higher than their measurements? Are there other studies supporting the yields?

10. Page 18, line 17. The Lee et al., 2016 PNAS study provide some constraints on isoprene nitrate yields and should be cited here.

11. Figures 9 and 10. What do the rest unlabeled ions represent?

12. Page 25, line 23. The authors provided evidence that MAE is not a product from MPAN oxidation, but then use MCM which assumes MAE has a 21% yield from MPAN oxidation, to estimate contribution to the 2MGA pathway. Why not modify the MCM model and provide an estimate that the authors are more confident about?

13. The CIMS data are shown for the LV pathway. CIMS can also measure HMML in the 2MGA pathway and I'm curious why the data were not shown.

---

## Author Response (AR1)

**Response to Review 1**

Thank you for the helpful comments and suggestions. We appreciate your time for reviewing our paper. We have addressed all of your comments as detailed below:

**p2 line 29: The authors may want to consider the publication of Peeters et al. PCCP 2014, if they mention new gas-phase chemistry of isoprene.**

Yes, good point, we had included the Wennberg et al., 2018 review paper as a summary of these advances in gas-phase chemistry, but to be more thorough we have added the following to include some examples of both recent and important experimental and theoretical studies. We assume you are referring to the Peeters et al. PCCP 2009 and Peeters et al. J. Phys. Chem. A., 2014 papers.

"Additionally, there have been major recent advances in our understanding of isoprene gas-phase oxidation (Wennberg et al., 2018, and references therein) including theoretical (e.g., Peeters et al., 2009, 2014; Kjaergaard et al., 2012) and experimental (e.g., Teng et al., 2017; Nguyen et al., 2015; Lee et al., 2014; Jacobs et al., 2014) studies. This improved understanding of isoprene gas-phase chemistry influences the processes governing isoprene SOA formation and informs the experimental design of the present work."

**p3, Fig 1: I would suggest to add reference for the chemical scheme that is shown. What is the importance of the 1,5-H shift reaction that is shown in the scheme for conditions of high NO in these experiments?**

Yes, we added a reference for both Figure 1 and Figure 2 and we removed the peroxy radical undergoing a 1,5-H shift to Figure 1. We do not detect the organic nitrate from this reaction, which is likely because this 1,5-H shift will not occur at the high levels of NO in these experiments. For clarity, we also add further detail describing Figure 1 in the introduction.

In the Figure 1 and 2 caption:

"largely based on schemes in Wennberg et al. (2018)"

And in the introduction:

"The formation of some organic nitrate SOA-precursors are summarized in Figure 1, which is largely adapted from schemes presented in Wennberg et al. (2018) with the exception of the isoprene dihydroxy nitrooxy alkoxy radical 1,5 H-shift. Wennberg et al. (2018) suggests the importance of a similar peroxy radical 1,5 H-shift, which will not form in the present experiments due to the high levels of NO. However, based on past studies largely on alkane oxidation (Orlando et al., 2003; Atkinson, 2007), the equivalent alkoxy radical 1,5 H-shift is expected to occur and has the potential to form low-volatility nitrates as further described in Section 5.1."

**p5 l6: I assume the specification of the Milli-Q water is meant to be 18 M Ohm.**

Yes, thanks we have updated this to "ultrapure water (18 M  $\Omega$ , Millipore Milli-Q)"

**p5 118: I would suggest to add the mean diameter of the seed aerosol for information.**

Yes, this has been added:

"The seed aerosol particle number concentration had an approximately lognormal diameter distribution centered on average  $\sim 100$  nm."

**p7 113: The author mention that in only few experiments NO2 was directly detected and in other NO2 was modelled. How was the model-measurement agreement of NO2 in the experiments, when NO2 was measured?**

Yes, unfortunately the luminol NO2/acyl peroxynitrate analyzer, which has less interferences than the Teledyne NOx analyzer was not operational for all experiments. In Figure S3, we show an example of a LV and 2MGA pathway experiment for modeled and measured NO2. The NO2 is reasonably well captured by the model, but as is explained in the supplement remaining biases are likely caused by measurement interferences or unaccounted for wall deposition of NOx reservoir species. We had added reference to this Figure in Section 3 of the main text, but we also now reference this Figure in the instrument section as well:

"The NO2 measured by the luminol NO2/acyl peroxynitrate analyzer compares reasonably well with the simulated NO2 from the kinetic model (Figure S3)."

**p8 l26: I would suggest to mention the parameters that were constrained by measurements in the model.**

Yes, this has been added:

"As listed in Table 1, the kinetic model was initialized for each experiment with the measured initial concentration of VOC, NO, NO2, and CH3ONO as well as the measured average temperature and relative humidity."

**p12 11-9: What does the unreasonable result of the correction of DMA data for wall loss mean for the uncertainty of yields determined for similar conditions in this work?**

The unreasonable result of the correction of DMA data described in this paragraph only applies to humid experiments and is why we do not report any DMA data from the humid experiments. We add the following to this paragraph to make this clearer.

"Thus, in this work, only the AMS results will be discussed for the humid experiments and SOA yields are only reported for experiments performed under dry conditions (Table 1). None of the dry experiments exhibited the odd behavior observed in the humid experiments, and the AMS results confirm that under dry conditions minimal nitric acid partitioned to the aerosols (Figure S10). For the dry experiments, the uncertainties are well characterized by the dry control experiments presented in Figure 3."

p14: Fig 5: The figure does not very clearly support the statement that SOA mass yield depends on the initial seed aerosol correction. There are essentially two values for the range from 1000 to 2000 and from 2500 to 6000 m2/cm3. Could you please

**comment?**

Yes, we have rephrased this in the text to be clearer and offer some further explanation. We interpret Figure 5 as having three regions: 0, 1000-1500, and 2500-6000 um2/cm3 instead of the two you specify. As seed aerosol increases, the SOA mass yield increases until a plateau is reached. This non-linear nature is expected especially for the isoprene SOA precursors as described below:

"Similar to previous studies (e.g., Zhang et al. (2014)), at a certain point increased seed surface area no longer substantially impacts the SOA yield (i.e., Figure 5 after 2500  $\mu$ m2 cm-3). This point will heavily depend on the system and the saturation mass concentration (C\*) of the SOA precursors. As shown in Table S2, the isoprene SOA precursors are mostly classified as IVOCs and SVOCs (Donahue et al., 2012). Reaching a point where most of the vapors are in particles relative to the chamber wall is expected for IVOCs and SVOCs, which have moderate vapor wall losses in Teflon chambers especially under dry conditions (Zhang et al., 2014; Huang et al., 2018)."

**p25 l27-30: Does the statement that SOA from the LV pathway is moderately higher than from 2MGA takes into account the differences in the turnover of the OH oxidation of the precursors of the two pathways?**

Yes, this is exactly why we do not directly compare the measured SOA yields from the LV and 2MGA pathways. As described in Section 5.5, we use the kinetic model at atmospheric conditions to estimate the contribution of both the LV and 2MGA pathways to the total. We have also added the following two sentences to make this clearer:

"In this study, direct comparison of the results from the 2MGA and LV pathways is difficult due to the difference in the extent of oxidation between the two regimes caused by the use of different VOC precursors and the variation in OH levels (Table 1). Thus, the kinetic model is used here to estimate the contribution of each pathway to the total under consistent oxidant levels."

**Response to Review 2**

Thank you for the helpful comments and suggestions. We appreciate your time for reviewing our paper. We have addressed all of your comments as detailed below:

Major: 1. The authors concluded in the abstract that the LV pathway which produces organic nitrates and dinitrates likely contribute to isoprene SOA under high-NO conditions more substantially than previously thought under typical atmospheric conditions. But some components of the typical atmospheric conditions were not thoroughly discussed. For example, how does hydrolysis of the organic nitrates affect this contribution? How would aerosol acidity affect the contribution from the 2MGA pathway? Would low particle surface areas in real atmosphere impede partitioning of the products that the authors discussed? In addition, as the authors have claimed themselves that the yields of these LV organic nitrates are highly uncertain, the MCM model assumes a yield of 10.4%. Estimation of LV pathway yield using this model is thus also highly uncertain. The results from this model that LV pathway yield of ~0.15 is rather hypothetic and I suggest leave it out of the abstract. Moreover, the partitioning behavior can be very different in the atmosphere compared to the chamber studies.

First, the 0.15 SOA yield is measured from the experiments (Figure 5), and not estimated from the kinetic model, so we leave this in the abstract, but revise the sentence as stated below:

"the experimentally measured SOA mass yield from the LV pathway is ~0.15"

We understand your concern that we too quickly extrapolate the impact of our results to the atmosphere. We agree that more detailed particle-phase box and regional/global modeling, which is out of scope of this work, would be necessary to fully understand the impact of our experimental results on SOA formed in the atmosphere, which we have already explained in Section 5.5. The modeling here is only meant to begin the process of comparing the SOA contribution from the two pathways. It is impossible to fully describe all of this in an abstract. Thus, we remove the rough attempt to approximate the contribution of each pathway to the atmosphere from the abstract and instead include a broader statement based only on our experimental results, which is the primary focus of this work. The simplified approximation is still included in Section 5.5 and in the conclusion, and we have included more explanation of the limitations of the simplified kinetic model used in this work in Section 5.5. As related to Major comment 2, this modeling is necessary in order to compare the two regimes under similar OH levels, from the same VOC precursor, and under atmospherically relevant conditions.

We change the last two sentences of the abstract to the following:

"The isoprene SOA mass yield from the LV pathway measured in this work is significantly higher than previous studies have reported suggesting that low-volatility compounds such as organic nitrates and dinitrates may contribute to isoprene SOA under high-NOx conditions significantly more than previously thought, and thus deserve continued study."

We add more detail in Section 5.5 and updated our approach to account for changes in the concentration of organic aerosol between chamber experiments and the ambient atmosphere. These revisions in our approach do not change the overall results significantly, but hopefully add more confidence:

"To estimate the aerosol contribution from the LV pathway, we assume that SOA production from the LV pathway scales with the production of isoprene dihydroxy dinitrates. Organic aerosol concentrations are higher in chamber experiments than the ambient atmosphere. By using low levels of VOC precursors compared to previous studies, this study attempts to reduce the organic aerosol concentrations to produce results more relevant to the ambient atmosphere. However, due to limitations in the DMA sensitivity, reducing the organic aerosol concentrations further to ambient levels is not possible. The ratio of the measured SOA yield (Figure 5) versus the simulated gas-phase dihydroxy dinitrate SOA precursor yield (Figure S4) is about 5. FP is decreased by a factor of 2 for the dihydroxy dinitrates when  $C_{OA}$  is reduced from ~25 µg cm-3 in the chamber to  $\sim 4\mu g$  cm-3 measured in the Southeast U.S. (Zhang et al., 2018). Thus, we multiply the dihydroxy dinitrate SOA precursors by 2.5 and we convert to mass by multiplying by the molecular weight of dihydroxy dinitrate. MCM v3.3.1 assumes a nitrate yield of 0.087-0.104 from NO reacting with the peroxy radical derived from OH + isoprene hydroxy nitrate. Low-volatility nitrates such as dihydroxy hydroperoxy nitrates form when HO2 reacts with the peroxy radical derived from OH + isoprene hydroxy nitrate. Such products would not form in the chamber conditions used in this work where NO levels remained above 100 ppb, but would form in the ambient atmosphere. Considering these low-volatility species from mixed chemical regimes would further increase the SOA mass generated from the LV pathway.

For the 2MGA pathway, we convert to mass by multiplying gas-phase HMML by the molecular weight of 2-MGA (120 g/mol), 2-MGA-nitrate (165 g/mol), and 2-MGA-sulfate (200 g/mol), which are the expected condensed-phase products under the high humidity levels in the atmosphere. Laboratory studies confirm that 2-MGA forms under humid conditions and some of the 2-MGA partitions to the gas-phase as expected based on its volatility (Nguyen et al., 2015). For simplicity, we assume most of the HMML forms 2-MGA-nitrate and 2-MGA-sulfate, but acknowledge further experimental and modeling studies are needed to fully understand HMML/2-MGA aqueous phase chemistry.

Then based on the gas-phase SOA precursor distribution from the kinetic model and assumptions above, under typical atmospheric conditions the fraction of the total SOA mass from isoprene OH-initiated oxidation under high-NOx conditions is ~0.7 from the LV pathway and ~0.3 from the 2MGA pathway. This assumes that the dihydroxy dinitrates are valid surrogates for the isoprene SOA. Considering many multi-functional isoprene derived organic nitrates have been detected in ambient aerosol (Lee et al., 2016), all SOA precursors in Table S2 with  $F_P > 0.05$  at 26°C are combined and converted to mass. Extrapolating these to ambient organic aerosol concentrations is more difficult because these compounds are more likely to exist in the particle phase because of accretion reactions and not volatility. When these products are assumed to exist entirely in the particle-phase and no factor is applied to correct for differences in organic aerosol

concentration or for these products only representing about 1/3 of the isoprene SOA yield measured in this study (Figure S4), the LV pathway is estimated to contribute to ~0.6 of the SOA formed under high-NOx conditions."

Also in Section 5.5, we added more detail explaining the need for additional experiments and modeling to more completely extrapolate the chamber SOA yields to the atmosphere, which is out of the scope of our current work.

"Additional studies addressing organic nitrate hydrolysis and aerosol acidity are also necessary to fully understand the relative impact of the two pathways on SOA formation. Additionally, the kinetic model used in this work only estimates gas-phase potential SOA precursors. Future analysis using a more complex model that explicitly simulates both the gas and particle phases would be useful for extrapolating the SOA yields measured here to the ambient atmosphere, which typically has lower organic aerosol concentrations than chamber experiments. This would need to be combined with additional analysis of the chemical constituents in the particle phase. From past work (Kleindienst et al., 2009; Xu et al., 2014; D'Ambro et al., 2017) demonstrating that isoprene derived SOA under high-NOx conditions is lower in volatility than that derived under low-NOx conditions and the C\* values estimated in this work (Table S2), accretion reactions appear to be important even in the LV pathway experiments. The degree to which accretion reactions occur in the LV pathway experiments to form even lower volatility products is quite uncertain and will greatly impact future analysis on how best to extrapolate isoprene SOA yields measured in chambers to the ambient atmosphere."

We also revise the concluding statement slightly:

"we now estimate based on the simple assumptions discussed in Section 5.5 that the LV pathway produces moderately more SOA mass than the 2MGA pathway due to the high isoprene SOA yield from the LV pathway measured in this work."

We also better explain why adding additional inorganic seed aerosol is necessary to compete with vapor wall losses in Section 4.2:

"With the addition of inorganic seed aerosol like ammonium sulfate, vapor species are expected to partition more to particles relative to the chamber wall (Zhang et al., 2014). The gas-particle equilibrium is not expected to be dependent on the concentration of inorganic seed aerosol, but instead is dependent on the concentration of organic aerosol. Depending on the saturation mass concentration (C\*), as the concentration of organic aerosol rises, vapors are present more in the particle-phase relative to the gas-phase (Seinfeld and Pandis, 2016). C\* and the fraction of a compound expected to be in the particle phase ( $F_P$ ) were estimated for a variety of organic nitrates and dinitrates in MCM v3.3.1 at 13, 26, and 32°C (Table S2)."

We would welcome future collaboration using the dataset described in this work to aid in development of regional and global models to more thoroughly determine the SOA contribution from the LV and 2MGA pathways. This is out of scope for the present work, which focuses on measurement of SOA yields, but we have added a statement in the Data Availability section to encourage future work in this direction:

"We welcome future collaboration with those who wish to use this data set for additional modeling purposes (e.g., creating volatility basis set parameters for global/regional models or for evaluating the results with a more complex box-model that includes aerosol chemistry). Please contact Rebecca Schwantes (rschwant@ucar.edu)."

2. Validation is needed to explain why the authors only studied the 2MGA pathway using methacrolein as the initial VOC. I understand that the authors tried to tailor the conditions to separate the two pathways and measure SOA yield from each pathway. This is a really good idea. But then, a direct comparison between the two pathways is also needed, which is more straightforward and reasonable to start with isoprene and vary NO/NO2 between different experiments. For example, using the many analytical tools in the authors' lab, they can show that as NO/NO2 decreases, the gas-phase (CIMS) and particle-phase (AMS) organic nitrates and dinitrates decrease, while the HMML and 2MGA increase. Comparing with the isolated experiments, quantification of the two pathways is probably possible too. In this way, the same OH oxidation extent and seed particle surface area will be used and variations can be better controlled. Although the 2MGA pathway from isoprene and methacrolein oxidation is the same mechanistically, in direction isoprene photooxidation experiment, the 2MGA pathway products are formed in later generation and that may change the dynamics (e.g., OH availability at different times) in the SOA formation. Also, how signature SOA product ions (both CIMS and AMS) from the LV pathway and the 2MGA pathway in the same isoprene photooxidation experiments as NO/NO2 ratio varies is a more desired and direct way to present the results.

Previous studies have already clearly demonstrated the impact of changing the NO2/NO ratio on the SOA yield from methacrolein and isoprene (e.g., Chan et al. 2010). The purpose of this study was not to repeat their results, but to attempt to completely separate the regimes and assess how the SOA from these two unique regimes respond to changing chamber conditions. From this we were able to understand more than past experiments. For example, if we had not completely separated the regimes, we would not have determined that low-volatility nitrates contribute appreciably to SOA formation as long as the experiments use high seed aerosol surface areas to limit the impact of vapor wall losses. We also would not have understood that the AMS is significantly more sensitive to aerosol from the 2MGA pathway and not the LV pathway. We potentially could have misinterpreted the results if we had only used the combined approach you describe above.

Unfortunately, the "combined" approach you describe above will result in mixed regimes throughout. The low-volatility nitrates from isoprene oxidation will form regardless of the NO2/NO ratio (see Figure 1 – none of these reactions are dependent on NO2). Even during methacrolein oxidation (Figure 2), organic nitrates are not formed in the competing pathway for the acyl peroxy radical reaction with NO versus NO2. So your statement that "as NO/NO2 decreases, the gas-phase (CIMS) and particle-phase (AMS) organic nitrates and dinitrates decrease, while the HMML and 2MGA increase" is not true. As NO/NO2 decreases, the organic nitrates from isoprene oxidation will largely be unaffected and HMML and 2MGA will increase. This is why we started with methacrolein as the precursor VOC for the 2MGA

pathway experiments. If we had started with isoprene, the 2MGA pathway experiments would have both nitrates and dinitrates from the LV pathway and 2MGA aerosol.

We add further description in Section 2 to explain this further:

"In order to completely separate the LV and 2MGA pathways, methacrolein had to be used as the VOC precursor for the 2MGA pathway experiments. If isoprene was used, even at the high NO2/NO ratios used in the 2MGA pathway experiments, the SOA precursors from the LV pathway would form resulting in a mixed regime (i.e., chemistry in Figure 1 is not dependent on NO2 concentration). In each case, the effect of seed surface area, temperature, and humidity on the SOA yield was independently determined."

You also suggest we use the CIMS and the AMS to differentiate the influence of these two pathways. Unfortunately, as we were performing these experiments, we realized that this was not possible due to limitations of both instruments. The CIMS can measure the organic nitrates, but we do not have sensitivities for many of these low-volatility nitrates to be quantitative. Additionally, the CF3O- CIMS cannot measure HMML. HMML, being a reactive alpha-lactone, is likely easily lost in the inlet or on sampling tubing and/or is not stable with the CF3O- ion chemistry. The AMS, which is most important for understanding the relative contribution of both pathways to SOA, is significantly more sensitive to aerosol from the 2MGA pathway than the LV pathway as explained in Section 5.3. Thus, the AMS cannot be used to separate the influence of the two pathways in mixed experiments as it realistically only measures aerosol from one of the regimes. As done here, the AMS can confirm that the two regimes are fully separated. So unfortunately, due to limitations in the CIMS and AMS techniques, we cannot separate the regimes as you suggest. You are correct that direct comparison to account for variation of OH and the extent of oxidation is important for comparing the two regimes. This is why we only use the kinetic model to directly compare the two regimes (Section 5.5) in this work. Additionally, based on the analysis of these chamber results, likely the best approach to improving isoprene SOA parameterizations in models is to parameterize the 2MGA and LV pathways separately. As explained further below, this work sets up the experimental basis for this approach. We add further description in Section 5.5 to explain this:

"This work was not only designed to independently study SOA formation from the two high-NOx regimes (the 2MGA and LV pathways), but also to suggest alternative methods for parameterizing isoprene SOA under high-NOx conditions in regional and global models. Because obtaining constant NO2/NO ratios similar to the ambient atmosphere is near impossible for a chamber study (e.g., temporal variation in Figure S3), creating isoprene SOA parameterizations based on NO2/NO ratio that realistically extrapolate to the ambient atmosphere is not realistic. Instead, this work highlights a potential alternative. Aerosol from the 2MGA pathway could be incorporated directly from gas-phase HMML formation and aerosol from the LV pathway could be included either from formation of surrogate compounds such as isoprene dihydroxy dinitrates or with a volatility basis set scheme. By treating the SOA from these two independent regimes separately, this study sets up the experimental basis for such an approach. In this study, direct comparison of the results from the 2MGA and LV pathways is difficult due to the difference in the extent of oxidation between the two regimes caused by the use of different VOC precursors and the variation in OH levels (Table 1). Thus, the kinetic model is used here to estimate the contribution of each pathway to the total under consistent oxidant levels."

Finally, the results reported here are novel because we were able to completely separate the LV and 2MGA pathways and test how varying chamber conditions impacted the two regimes independently. These novel results were missed by other past studies, which took the "combined" approach you suggest. We have added further explanation of this in Section 5.4:

"As shown in Table 2, a variety of  $NO_x$  regimes (i.e., non-consistent  $NO_2/NO$  ratios) are all labeled as high- $NO_x$  in these past studies. Each study likely produces SOA in varying degrees from the LV and 2MGA pathways, which greatly complicates direct comparison between these past studies. By varying a large number of conditions and completely separating SOA production between the 2MGA and LV pathways, our results lend insight into the variation in these past experiments."

3. Page 12, line 22. It is very interesting that SOA yield does not vary too much as temperature changes between 13 and 32 C. The authors mention two reconciling effects: enhanced vapor wall loss and enhanced organic nitrate yields from RO2 + NO reaction under lower temperatures. Can the authors provide some quantitative constraints on these two effects? How different are the organic nitrate yields and wall loss rates under this range of temperatures? Another point to add, is that at lower temperature, more semivolatile species may partition into the particle phase. I think this is a major effect. Previous studies using very similar temperature range and found a difference of 2-10 times in SOA yields (Clark et al., 2016 ES&T; Takekawa et al., 2003 Atmos. Environ.; Sheehan and Bowman, 2001 ES&T.). Some of these studies use isoprene, some use hydrocarbons that likely produce less volatile SOA than isoprene. The authors should think about addressing this question. Is it because the LV SOA have surprisingly low volatility that they are already mostly in the particle-phase at 30C? Can the authors estimate the vapor pressure of the nitrate and dinitrate compounds and see if this hypothesis make sense? Note that there are only two experiments at different temperatures and the lower temperature experiment started forming SOA much earlier than the higher temperature experiment. But they ended up forming similar SOA mass. From that perspective, there is a clear temperature influence. I hope the authors can address this question with more detailed discussion. The same case for the 2MGA pathway. The authors found little temperature influence on SOA mass yield. But again, that was based on only one high temperature and one low temperature experiment. The two experiments have very different seed particle surface areas too. The data provided are not sufficient to draw conclusions on temperature effects.

Thank you for pointing out the Clark et al., 2016 ES&T paper. We have now included this in Table 3 (now Table 2) and added a column for chamber volume. We explain how our results compare to this paper in Section 4.2, 4.3 and 5.4. Given that the SOA yields measured from experiments at 13 and 32 deg C were not significantly different for the LV or 2MGA pathway, we chose not to perform additional temperature experiments. Unfortunately, 13 deg C is near the

limit of the Caltech chamber's temperature capability, so lower temperature experiments are not possible. Thank you for the recommendation of estimating the C\* values. We have now estimated the C\* and estimated fraction of a compound in the particle phase (FP) at 13, 29, and 32°C (Table S2) for all organic nitrates and dinitrates from MCM v3.3.1 relevant to the LV pathway. We include a description of how these C\* and FP values were calculated in Section S1 of the supplement and briefly in Section 3. We also include substantial updates as explained below referencing these C\* values and addressing your concerns above:

**Explanation of C\* values in Section 3:**

[revised manuscript text omitted]

We also update Section 4.3 and update Figure 7. In Figure 7, two of the experiments have nearly identical seed surface areas. For ease of viewing we had shifted one over. We have reduced the size of this shift. The seed surface areas for the three temperature experiments (13, 26, and 32°C) are within 600 um2/cm3 of each other. This seems sufficient to demonstrate that temperature does not greatly impact the SOA yield in the 2MGA pathway at high NO2/NO levels.

"At the high NO2/NO ratios used in this work, temperature does not impact SOA mass yield beyond given uncertainties (Figure 7). Based on known gas-phase chemistry, past studies (e.g., Clark et al. (2016)) with more moderate NO2/NO ratios than that used in this work are expected to measure an enhanced SOA yield under colder temperatures due to a reduction in MPAN thermal decomposition and thereby an increase in HMML formation."

We also update Section 5.4:

"As shown in Table 2, the range for isoprene SOA yields under high-NOx conditions even from the two most recent studies at comparable temperatures spans over an order of magnitude (0.004 at ~21°C for Bregonzio-Rozier et al. (2015) and 0.1 at 27°C for Clark et al. (2016)). Our results are most consistent with those of Clark et al. (2016). As shown in Table 2, a variety of NOx regimes (i.e., non-consistent NO2/NO ratios) are all labeled as high-NOx in these past studies. Each study likely produces SOA in varying degrees from the LV and 2MGA pathways, which greatly complicates direct comparison between these past studies. By varying a large number of conditions and completely separating SOA production between the 2MGA and LV pathways, our results lend insight into the variation in these past experiments.

Many of the past SOA yield measurements were performed with no seed aerosol. Consistent with past results, when no seed aerosol was injected into the chamber (experiments D1 and M1), the SOA mass yield for the LV pathway (0 from isoprene) and 2MGA pathway (0.1 from methacrolein) were quite low. Past experiments performed with no seed aerosol were only measuring SOA from the 2MGA pathway, which is highly dependent on the NO2/NO ratio (Chan et al., 2010), which varied greatly between these past studies. Clark et al. (2016), who measured high SOA yields (0.1 at 27°C) in unseeded experiments is the exception. Possibly, the larger chamber volume (90 m3) used by Clark et al. (2016) compared to most studies listed in Table 2 reduced vapor wall losses and contributed to the enhanced SOA yield. However, other chamber characteristics might also be important because Zhang et al. (2011) measured quite low isoprene SOA yields (0.007-0.03) using a chamber larger than the one used in the Clark et al. (2016) study.

While the zero or low seed aerosol loading experiments in this study generally compare well with the past, SOA yields measured here using higher initial seed surface areas are substantially greater than most studies, especially for the LV pathway. The SOA yield from the LV pathway is ~0.15 in this study, while past isoprene SOA yields are largely ~0.07 with the exception of studies optimizing for high RO2 + NO2 reactions (Chan et al., 2010) or mixed regimes - RO2 + HO2/NO (Xu et al., 2014). The SOA yield from the LV pathway in this work is even larger than the SOA yield from Clark et al. (2016) (0.1 at 27°C), which includes SOA from both the LV and 2MGA pathways. Possibly the larger chamber volume used by Clark et al. (2016) reduces vapor wall losses, but not to the extent that enhanced seed surface area does in this work."

**4. The authors use "low-volatility" to describe the organic nitrates. But they should also justify this by providing some estimates of the vapor pressures or C\* of the molecules in Figure 1.**

As explained in Major comment 3, we have estimated C\* values for relevant organic nitrates and dinitrates in MCM v3.3.1. These are listed now in Table S2. In the introduction, we also define how we use low-volatility throughout the text:

"Throughout the text we use low-volatility as a general term representing gas-phase compounds with a potential to exist partially in the particle phase. In this work, lowvolatility compounds include the following volatility classes from Donahue et al. (2012): IVOC (Intermediate), SVOC (Semi-), LVOC (Low), and ELVOC (Extremely Low). When referring to specific volatility classes, the acronyms defined above are used."

Minor: 1. Title and abstract: HMML is formed not only under high-NO, but also high-NO2 (MPAN is their precursor). It is thus inaccurate to term "high-NO" as the condition. As the authors also point out in the text, NO2/NO ratio is an important indicator of the two pathways. From the experimental list, NO and NO2 are indeed adjusted to optimize

**different conditions. Thus, it seems more appropriate to use "high NOx" rather than "high-NO" in this manuscript.**

We had chosen high-NO here based only on the first step of isoprene oxidation by OH. This peroxy radical will not react with NO2. This then defines the scope of our work, which is all subsequent chemistry past the isoprene hydroxy peroxy radical + NO reaction. However, we do understand your more general interpretation too, which is based on all peroxy and acyl peroxy radicals in the system reacting with either NO or NO2.

Thus, we change all instances of high-NO to high-NOx in the paper unless we are particularly referring to only high-NO or high-NO2 conditions, and we also add further explanation in the introduction:

"There are many definitions for high-NOx conditions (Wennberg, 2013). Here we test two different high-NOx chemical regimes. Experiments targeting the LV pathway are designed such that all peroxy radicals including acyl peroxy radicals dominantly react with NO and experiments targeting the 2MGA pathway are designed such that all acyl peroxy radicals dominantly react with NO2 and all other peroxy radicals dominantly react with NO."

**2. Page 3, line 2. The 2MGA pathway should refer to particle-phase products (2-MGA, its oligomers, and its organosulfates) from further oxidation of MPAN or uptake of HMML.**

Yes, we updated this as suggested to include more detail:

"representing aerosol formed from 2MGA, its oligomers, its organosulfates, and its organonitrates (blue compounds in Figure 2)"

**3.** Page 5, line 27. Whether the NO2 signal can be interferences from nitrous acid or CH3ONO can be easily tested using the standards.**

Yes, we know CH3ONO and nitrous acid are interferences based on standards. As suggested we adjust the text to be more descriptive:

"(e.g., known interferences include organic nitrates, nitrous acid, and CH3ONO)."

**4. Page 12, line 14. It is unclear what is referred to as "the 2-MGA precursor" here. Is it MACR? MPAN? HMML? This information was obtained from the kinetic mechanism. I think the author should at least provide the simulations in the supporting information and mention it in the main text.**

We understand your confusion and have completely revised this so that the use of the kinetic mechanism is more straightforward. Originally, we had added "tracers" for both the 2MGA (HMML + MAE) and LV pathway (dihydroxy dinitrates) into the mechanism. Given your comment, we have decided to change this. We have made some small changes to the MCM v3.3.1 mechanism to be more consistent with recommendations from the Caltech isoprene mechanism (Wennberg et al. 2018). We have listed all changes to MCMv3.3.1 in the original Table 2, which has been moved to the supplement (now Table S1) due to length. We have added text summarizing these changes in Section 3:

"Updates include inorganic reactions needed for chamber studies with large NOx levels (e.g., CH3ONO photolysis) and small changes to the isoprene chemistry based largely on Wennberg et al. (2018) and consistent with Figures 1 and 2. As shown in Table S1, these updates include: the first generation isoprene hydroxy nitrate yields, the rates and branching ratios for the oxidation of the first generation isoprene hydroxy nitrates, and the HMML yield from the MPAN + OH reaction. In some cases,  $\delta$ -isoprene hydroxy alkoxy radicals in MCM v3.3.1 decompose through peroxy radical H-shifts directly to products that would not form under the high-NO conditions in this work. For simplicity, we change these reactions, so that the  $\delta$ -isoprene hydroxy alkoxy radicals form unity yields of hydroxy aldehydes."

Now that we have directly updated HMML formation in the mechanism, we can refer to HMML rather than the 2-MGA precursor throughout the text. We have updated this in Section 4.2, Section 4.3, Section 5.5, and in S1 of the Supplement.

**5. Page 12, line 26-27. Exceptions are when temperature was low and initial seed particle surface area was high.**

We have already explained that the low temperature experiment forms aerosol earlier (see major comment 3). Here we add an explanation for the higher seed surface area experiments as well: "As shown in Figure 4, generally, SOA formation begins earlier (i.e., with less isoprene reacted) in experiments with larger seed aerosol. This is consistent with vapors partitioning more to particles relative to the chamber wall when seed aerosol is

enhanced."

6. Page 12, line 31. Based on the earlier description, the kinetic model combines MCM and some other gas-phase reactions that are listed in Table 2. So the kinetic model seems to be a pure gas-phase model. It is thus unclear when the authors mentioned that the kinetic model can predict the SOA mass yield. This sentence needs to be re-phrased to something like: "The kinetic model predicts that the formation of gas-phase dinitrates from experiment D7 is similar to the other experiments". Then the followed issue is whether this single species can represent the overall SOA in the studied condition. The authors may want to also compare other proposed LV-pathway products from the model.

Yes, we revised this sentence:

"The kinetic model predicts that the production of important gas-phase SOA precursors from the LV pathway (e.g., isoprene dihydroxy dinitrates), when corrected for total isoprene reacted, is similar in experiment D7 to the other experiments (Figure S4)."

We now discuss how other potential SOA precursors may also influence the isoprene SOA yield in Section 4.2 (see response to major comment 3). Figure S4 has also been updated. Unfortunately, use of the dihydroxy dinitrates as a surrogate for SOA from the LV pathway is necessary because the exact composition is unknown. We add to section 5.5, another approach using all of the compounds in Table S2 with  $F_P > 0.05$  at 26°C. This produces similar results to using dihydroxy dinitrates as the surrogate for LV pathway SOA and hopefully adds confidence to this approach. "This assumes that the dihydroxy dinitrates are valid surrogates for the isoprene SOA. Considering many multi-functional isoprene derived organic nitrates have been detected in ambient aerosol (Lee et al., 2016), all SOA precursors in Table S2 with  $F_P > 0.05$  at 26°C are combined and converted to mass. Extrapolating these to ambient organic aerosol concentrations is more difficult because these compounds are more likely to exist in the particle phase because of accretion reactions and not volatility. When these products are assumed to exist entirely in the particle-phase and no factor is applied to correct for differences in organic aerosol concentration or for these products only representing about 1/3 of the isoprene SOA yield measured in this study (Figure S4), the LV pathway is estimated to contribute to ~0.6 of the SOA formed under high-NOx conditions."

**7. Page 15, line 9. The authors may also want to consider that organosulfate formation in the presence of ammonium sulfate seed particles.**

Yes, we have added the following sentence to account for this:

"Alternatively, the presence of higher ammonium sulfate seed aerosol may also increase organosulfate formation, which could impact SOA composition and yield."

**8. Page 15, line 15. HMML/reacted MACR = 0.25 does not mean the SOA yield from HMML uptake is 0.25. HMML is a C4H6O3 lactone. If you think about its vapor pressure, it is probably on the more volatile side of the SVOC range. It will have a large fraction in the gas phase.**

Yes, we have now added in more detail here:

"HMML, based on volatility alone, would exist mostly in the gas phase, but because HMML is very reactive (e.g., oligomerization or reaction with inorganic ions in the particle phase), HMML quickly produces aerosol (Kjaergaard et al., 2012; Nguyen et al., 2015). Based on HMML production simulated by the kinetic mechanism under the conditions used in these experiments, ~0.21 SOA mass yield from methacrolein is expected purely from the mass contained in HMML (MW = 102 g/mol, Figure S4). At first, the molecular weight of HMML itself is used because this is the mass of the majority of the oligomer monomers."

**9. Page 17, line 30 to Page 18, line 2. The description is unclear. A brief but better explanation (better than just saying "highly uncertain") is needed here to explain why the Lee et al. used upper limit yields 10 times higher than their measurements? Are there other studies supporting the yields?**

Lee et al. 2014 based these uncertainties on carbon closure. We recognize that this might not be the best approach especially considering that the uncertainties in the sensitivities of the measured compounds could also explain the lack of carbon closure (Lee et al., 2014, Interactive comment on "Kinetics of the reactions of isoprene-derived hydroxynitrates: gas phase epoxide formation and solution phase hydrolysis" by M. I. Jacobs et al.). Thus, we revise this sentence to be more general:

"One study, Lee et al. (2014), was able to quantify the yield of dinitrates from the firstgeneration isoprene hydroxy nitrate standards. Assuming a sensitivity similar to the isoprene hydroxy nitrate standards, Lee et al. (2014) measured a dinitrate yield of 0.03-0.04 from OH-initiated oxidation of the  $\delta$ -1-hydroxy,4-nitrate isomer."

**10. Page 18, line 17. The Lee et al., 2016 PNAS study provide some constraints on isoprene nitrate yields and should be cited here.**

Yes, we had already cited this paper in Section 5.3 and the conclusions, but we now cite this paper here too:

"Many multi-functional isoprene derived organic nitrates have been detected in ambient aerosol (Lee et al., 2016)."

**11. Figures 9 and 10. What do the rest unlabeled ions represent?**

We add "in gray" to the description in Figure 9 and 10 to clarify that all of the mass spectra are in gray and then specific m/z are labeled in the various colors. The aerosol mass spectrometer produces spectra with a high degree of fragmentation. Thus, as has been done by previous studies on isoprene oxidation (e.g., Chan et al. 2010 for 2-MGA and 2-MGA oligomers aerosol and Lin et al. (2012) for IEPOX aerosol), we only characterize the ions that form distinctly high peaks.

**12. Page 25, line 23. The authors provided evidence that MAE is not a product from MPAN oxidation, but then use MCM which assumes MAE has a 21% yield from MPAN oxidation, to estimate contribution to the 2MGA pathway. Why not modify the MCM model and provide an estimate that the authors are more confident about?**

We have now modified MCM v3.3.1, see Table S1 for a list of reactions and a description of these changes in Section 3. See the response to minor comment 4 for more detail.

**13. The CIMS data are shown for the LV pathway. CIMS can also measure HMML in the 2MGA pathway and I'm curious why the data were not shown.**

Unfortunately, the  $CF_3O^-$  CIMS, used in this work, cannot measure HMML. HMML, being a reactive alpha-lactone, is likely easily lost in the inlet or on sampling tubing and/or is not stable with the  $CF_3O^-$  ion chemistry.

**Low-volatility compounds contribute significantly to isoprene SOA under high-NOx conditions**

Rebecca H. Schwantes1,5, Sophia M. Charan2, Kelvin H. Bates2,6, Yuanlong Huang1, Tran B. Nguyen3, Huajun Mai1, Weimeng Kong2, Richard C. Flagan2, and John H. Seinfeld2,4

[revised manuscript text omitted]